# Oxygen enhances antiviral innate immunity through maintenance of EGLN1-catalyzed proline hydroxylation of IRF3

Xing Liu[1,2,3,4,6], Jinhua Tang[1,5,6], Zixuan Wang[1,3,6], Chunchun Zhu[1,3], Hongyan Deng [1], Xueyi Sun[1,3], Guangqing Yu[1,3], Fangjing Rong[1,3], Xiaoyun Chen[1,3], Qian Liao[1,3], Shuke Jia[1,3], Wen Liu[1,3], Huangyuan Zha[1], Sijia Fan[1], Xiaolian Cai[1], Jian-Fang Gui [1,2,3,4] & Wuhan Xiao [1,2,3,4] ✉

Oxygen is essential for aerobic organisms, but little is known about its role in antiviral immunity. Here, we report that during responses to viral infection, hypoxic conditions repress antiviral-responsive genes independently of HIF signaling. EGLN1 is identified as a key mediator of the oxygen enhancement of antiviral innate immune responses. Under sufficient oxygen conditions, EGLN1 retains its prolyl hydroxylase activity to catalyze the hydroxylation of IRF3 at proline 10. This modification enhances IRF3 phosphorylation, dimerization and nuclear translocation, leading to subsequent IRF3 activation. Furthermore, mice and zebrafish with Egln1 deletion, treatment with the EGLN inhibitor FG4592, or mice carrying an Irf3 P10A mutation are more susceptible to viral infections. These findings not only reveal a direct link between oxygen and antiviral responses, but also provide insight into the mechanisms by which oxygen regulates innate immunity.

Among the essential factors required for the life of aerobic organisms, oxygen ($O_2$) is indispensable due to its critical role in the generation of energy, metabolites, and structural macromolecules[1,2]. In addition, $O_2$ also acts as a direct substrate for some enzymes, such as the EGLN prolyl hydroxylases (EGLN1, EGLN2, and EGLN3), the factor inhibiting HIF asparaginyl hydroxylase, the JmjC domain histone demethylases, and the TET methylcytosine oxidases[1,3–5]. To adapt to variable oxygen tensions, aerobic organisms have evolved multiple mechanisms to sense $O_2$ levels, such as through Fe (II)/2-OG-dependent oxygenases (e.g., EGLN1-3, FIH and KDM6A), and thiol dioxygenases (e.g., 2-aminoethanethiol dioxygenase), thereby regulating $O_2$ consumption and physiological processes[1,2,6–8]. Among of them, the EGLN-HIF $O_2$-sensing system is the most critical oxygen monitoring machinery used by metazoans, and the EGLNs (EGLN1, EGLN2, and EGLN3) serve as oxygen sensors[2,9–11].

Accumulating evidence supports a bidirectional relationship between viral replication and tissue oxygen levels. In general, the replication of some viruses is enhanced in infected tissues under low-oxygen conditions, whereas some viruses have a selective tropism for high-oxygen environments[12–14]. In contrast, hypoxia limits the replication of some viruses that naturally infect organs with high oxygen tension and induces the replication of some viruses that naturally infect organs with low oxygen tension[13,14]. Because oxygen is a central component of the oxidative metabolism of all tissues, either directly or indirectly, and there is a critical relationship between metabolic changes and immune responses, oxygen tension is a recognized pillar of the inflammatory and immune responses[15,16]. In particular, the role of hypoxia in immune responses remains controversial. On one hand, hypoxia is a common and prominent microenvironmental feature at sites where the innate immune system is activated[17–19]. On the other hand, hypoxia has been shown to create an immunosuppressive environment within the tumor mass, leading to immune evasion[20–22]. As a master regulator of the cellular responses to $O_2$, the hypoxia-

[1]Key Laboratory of Breeding Biotechnology and Sustainable Aquaculture, Institute of Hydrobiology, Chinese Academy of Sciences, Wuhan, P. R. China. [2]Hubei Hongshan Laboratory, Wuhan, P. R. China. [3]University of Chinese Academy of Sciences, Beijing, P. R. China. [4]The Innovation Academy of Seed Design, Chinese Academy of Sciences, Wuhan, P. R. China. [5]Department of Pharmacy, Women and Children's Hospital of Chongqing Medical University, Chongqing, P. R. China. [6]These authors contributed equally: Xing Liu, Jinhua Tang, Zixuan Wang. ✉e-mail: w-xiao@ihb.ac.cn

inducible factor (HIF) has been shown to play a prominent role in the regulation of innate immunity[15,16,23,24], but it appears that hypoxia-induced downregulation of the type I IFN pathway is independent of HIFα[25]. During immunity and inflammation, HIFα stabilization occurs, which may directly regulate the expression of immune genes[23]. Certain viruses have even evolved mechanisms to directly stabilize HIF1α in order to exert an anti-apoptotic effect and prolong the survival of the infected cells[24]. To date, most of our understanding of the role of oxygen in innate immunity is related to the HIF signaling pathway. However, the direct role of oxygen in innate immunity and the underlying mechanisms remain elusive. Despite the existing controversy between tissue oxygen status and viral propagation[12–14], the benefits of oxygen on host antiviral immune responses have been recognized[14,15].

To defend against viral infection, innate immunity serves as the first line of defense, detecting RNA and DNA viruses and subsequently triggering host antiviral responses, such as the production of type I interferons and pro-inflammatory cytokines, leading to the clearance of invading viruses or the onset of inflammatory responses[26–28]. Interferon regulatory factor 3 (IRF3), a key transcription factor responsible for the induction of IFNs, is constitutively expressed in cells and is key to the rapid release of antiviral molecules localized to the cytoplasm of resting cells[29]. Upon pathogen recognition, IRF3 is phosphorylated on multiple serine and threonine residues by the upstream kinases tank-binding kinase 1 (TBK1) and inhibitor-κB kinase ε (IKKε). Phosphorylated IRF3 then undergoes dimerization, leading to the translocation of IRF3 dimers from the cytoplasm to the nucleus, where IRF3 activates the expression of type I IFNs and subsequent interferon-stimulated genes (ISGs)[30,31]. IRF3 is tightly regulated by post-translational modifications (PTMs), including phosphorylation, ubiquitination, SUMOylation, S-glutathionylation, acetylation, methylation, dephosphorylation and de-ubiquitination[32–42]. However, whether hydroxylation regulates IRF3 function in antiviral immunity and whether crosstalk between hydroxylation and conventional PTMs orchestrates this process remains largely unknown.

Here, we show that the antiviral immune response genes are repressed under hypoxia in response to viral infection independently of HIF signaling. EGLN1 positively regulates antiviral immune responses by binding to IRF3 and catalyzing IRF3 hydroxylation at proline 10. Disruption of *Egln1* in mice and zebrafish attenuates host innate antiviral immune responses, and Irf3 prolyl hydroxylation-deficient mice are more susceptible to lethal viral infections.

## Results

### Oxygen enhances cellular antiviral immune responses

We had previously observed that the survival rate of virus-infected zebrafish was greatly decreased after reducing oxygen concentration in their living water[43]. This prompted us to investigate whether oxygen has an effect on antiviral immunity. First, we compared the expression of the antiviral *IFN-β* gene and the antiviral IFN-stimulated gene *CXCL10*, following viral infection under normoxia and hypoxia[44,45]. When cells of the human non-small cell lung carcinoma H1299 line were treated under hypoxia (1% $O_2$), HIF1α protein was stabilized (Supplementary Fig. 1a), and the typical hypoxia-responsive genes, including *PDK1, GLUT1, LDHA, VEGF* and *BNIP3*, were largely upregulated[46], indicating that the hypoxic condition was achieved (Supplementary Fig. 1b). In response to vesicular stomatitis virus (VSV) infection, the induced expression of *IFN-β* and *CXCL10* was significantly reduced under hypoxia compared to normoxia (Fig. 1a). Consistent with these findings, in response to transfection with increasing amounts of TLR-3 ligand, poly (I:C), the induced expression of *IFN-β* and *CXCL10* was also reduced under hypoxia compared to normoxia (Fig. 1b).

HIF1α plays an important role in interferon production[47]. To exclude the effect of HIF signaling on antiviral gene expression under

hypoxia, we knocked out *HIF1β* (*ARNT*), a component of the HIFα/β heterodimer required for HIF signaling activation[46,48], in H1299 cells (Supplementary Fig. 1c)[49,50]. As expected, the expression of *LDHA* and *VEGF* [46], was unable to be induced under hypoxia (Supplementary Fig. 1d), demonstrating the effective inactivation of HIF signaling in *HIF1β*-deficient H1299 cells. After VSV infection with increasing doses, the induced expression of *IFN-β* and *CXCL10* was still reduced under hypoxia compared to normoxia (Fig. 1c). Furthermore, we used PX478, a specific HIF1α inhibitor[51,52], to block HIF1α activity in H1299 cells and confirmed these observations. As expected, the expression of *GLUT1*[46], was unable to be induced under hypoxia (Supplementary Fig. 1e), confirming the effective inhibition of HIF1α in H1299 cells. After VSV infection, the expression of *IFN-β* and *CXCL10* was still reduced under hypoxia compared to normoxia (Fig. 1d). Similar results were obtained using the human monocytic cell line THP-1 (Supplementary Fig. 1f, g and Fig. 1e). Furthermore, IFN-β protein levels were significantly reduced in THP-1 cells under hypoxia compared to normoxia upon herpes simplex virus 1 (HSV-1) infection (Fig. 1f).

Under the physiological conditions, the oxygen concentrations in human tissues are rarely as high as 21% (≈160 mmHg) and less than 1% (≈7 mmHg)[53,54]. To determine whether our finding that oxygen affects innate antiviral responses is applicable under physiological oxygen conditions, we used three oxygen pressures for further comparative assays, including 7 mmHg (≈1% $O_2$) (normally found in the skin epidermis and represented as the lowest oxygen pressure in human tissues)[55], 20 mmHg (≈2.7% $O_2$) (represented as one of the lower oxygen pressures in human tissues)[54], and 160 mmHg (21% $O_2$) (air oxygen concentration; normally treated as normoxia control). When THP-1 cells were treated under 160 mmHg, 20 mmHg or 7 mmHg $O_2$, HIF1α protein increased steadily as the oxygen concentration decreased (Fig. 1g). Upon challenge with VSV or HSV-1, the expression of *IFN-β* and *CXCL10* was also steadily decreased in response to the decrease in oxygen concentration (Fig. 1h).

As H1299 and THP-1 are cancer-derived cell lines, we sought to determine whether the effect of oxygen on innate antiviral responses was also present in primary human immune cells. To this end, we performed further assays in peripheral blood mononuclear cells (PBMC) and leukocytes. When PBMCs were treated under hypoxia (1% $O_2$), the expression of *PDK1, LDHA*, and *VEGF*, was upregulated (Supplementary Fig. 1h). After VSV infection, the expression of *IFN-β, ISG15*, and *IFIT1* was reduced under hypoxia compared to normoxia (Supplementary Fig. 1j). Similar results were obtained in leukocyte cells (Supplementary Fig. 1i, k).

Taken together, these data suggest that oxygen promotes innate antiviral responses in immune and non-immune cells independently of HIF signaling.

### *EGLN1* positively regulates cellular antiviral immune responses depending on its enzymatic activity

*EGLN1* is a key oxygen sensor that transduces oxygen signaling to downstream events by catalyzing proline hydroxylation of its targets[46,56]. To determine whether the enhancement of antiviral responses under normoxia is mediated by *EGLN1*, we examined the effect of *EGLN1* on immune responses upon viral infection. Upon infection with Sendai virus (SeV), ectopic overexpression of *EGLN1* significantly induced the expression of *IFN-β, CXCL10*, and *CCL5* in HEK293T cells (Supplementary Fig. 2a, b). Conversely, the expression of *IFN-β* and *ISG15* in *EGLN1*-deficient HEK293T cells (*EGLN1*$^{-/-}$) was much lower than *EGLN1*-sufficient HEK293T cells (*EGLN1*$^{+/+}$) upon SeV infection (Supplementary Fig. 2c, d).

Ectopic expression of *EGLN1* also caused an increase of *IFN-β* and *CXCL10* expression in H1299 cells upon VSV infection (Supplementary Fig. 2e, f). Consistent with these findings, ectopic expression of *EGLN1* caused a reduction of VSV-GFP positive cells in H1299 cells (Supplementary Fig. 2g). In response to SeV, VSV or HSV-1 infection, the

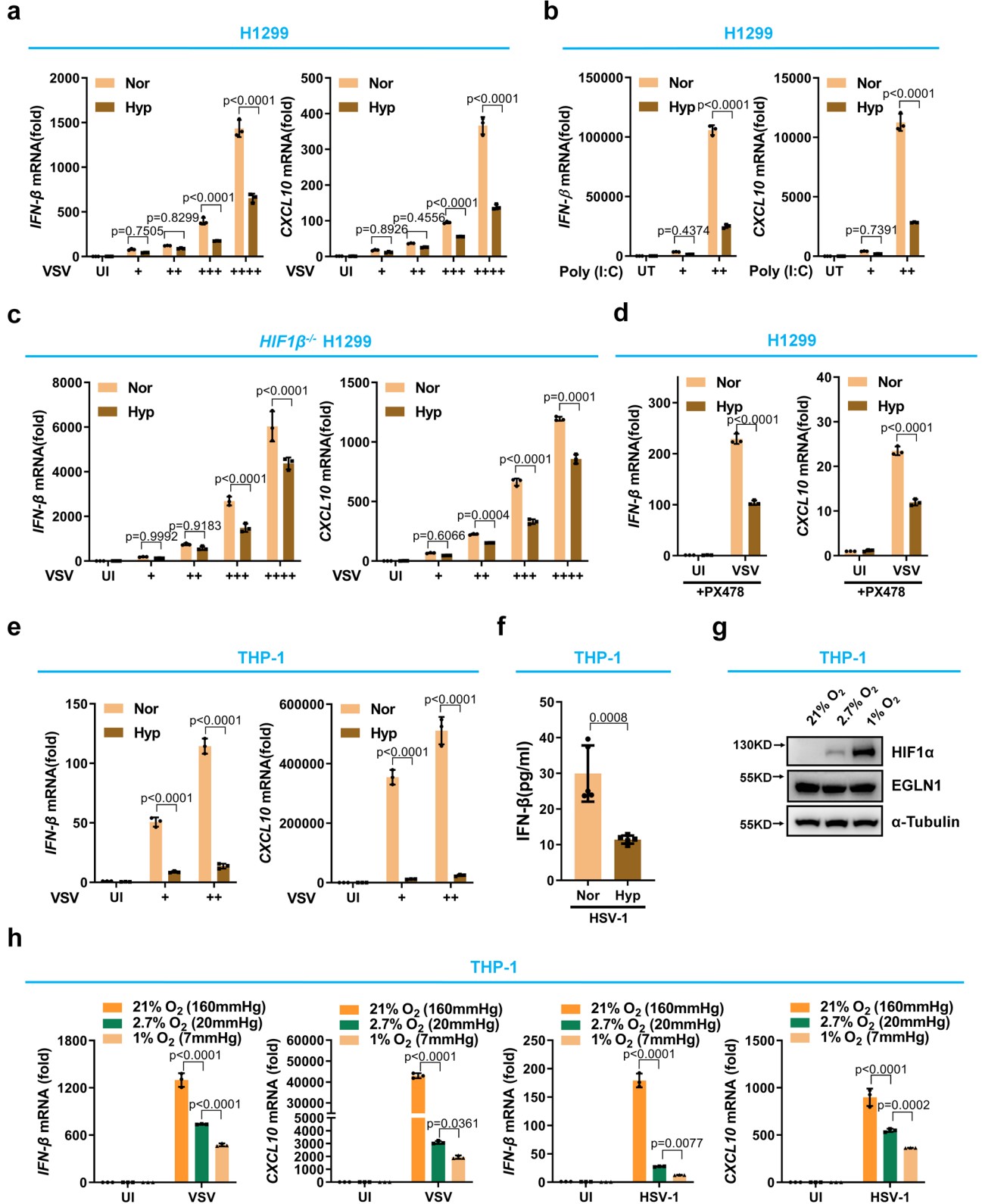

expression of *IFN-β*, *CCL5* or *CXCL10* was reduced in *EGLN1*⁻/⁻ H1299 cells compared to *EGLN1*⁺/⁺ H1299 cells (Fig. 2a and Supplementary Fig. 2h, i). Consistent with these findings VSV-GFP positive cells were more abundant in *EGLN1*⁻/⁻ H1299 cells than in *EGLN1*⁺/⁺ H1299 cells (Fig. 2b). Similarly, in response to SeV, VSV, or HSV-1 infection, *IFN-β* expression was significantly reduced in *EGLN1*⁻/⁻ THP-1 cells compared to *EGLN1*⁺/⁺ THP-1 cells (Supplementary Fig. 2j and Fig. 2c).

To determine whether the effect of *EGLN1* on antiviral immune responses is dependent of *VHL*[1], we used two clear cell renal cell carcinoma cell lines, RCC4 (VHL-deficient; with both intact HIF1α and intact HIF2α) and 786-O (VHL-deficient; with intact HIF2α but deficient HIF1α)[57,58]. In response to VSV or HSV-1 infection, the expression of *IFN-β* was lower in *EGLN1*⁻/⁻ RCC4 and *EGLN1*⁻/⁻ 786-O cells than in *EGLN1*⁺/⁺ RCC4 and *EGLN1*⁺/⁺ 786-O cells (Supplementary Fig. 2k, l and Fig. 2d, e),

**Fig. 1 | Hypoxia suppresses antiviral gene expression in response to viral infection independently of HIF signaling. a** qPCR analysis of *IFN-β* and *CXCL10* mRNA in H1299 under normoxia (21% O₂) or hypoxia (1% O₂), infected without (UI) or with increasing amounts of VSV (+, 0.2 MOI (Multiplicity of Infection); ++, 0.5 MOI; +++, 1.0 MOI; ++++, 2.0 MOI) for 8 h. **b** qPCR analysis of *IFN-β* and *CXCL10* mRNA in H1299 under normoxia (21% O₂) or hypoxia (1% O₂), transfected without (UT) or with increasing amounts of poly (I:C) (+, 0.5 μg/ml; ++, 1.0 μg/ml) for 8 h. **c** qPCR analysis of *IFN-β* and *CXCL10* mRNA in *HIF1β⁻/⁻* H1299 under normoxia (21% O₂) or hypoxia (1% O₂), infected without or with increasing amounts of VSV (+, 0.2 MOI; ++, 0.5 MOI; +++, 1.0 MOI; ++++, 2.0 MOI) for 8 h. **d** qPCR analysis of *IFN-β* and *CXCL10* mRNA in H1299 treated with PX478 (10 μM) under normoxia (21% O₂) or hypoxia (1% O₂), infected without or with VSV for 8 h. **e** qPCR analysis of *IFN-β* and *CXCL10* mRNA in THP-1 under normoxia (21% O₂) or hypoxia (1% O₂), infected without or with increasing amounts of VSV (+, 1.0 MOI; ++, 2.0 MOI) for 8 h. **f** ELISA assay of IFN-β in THP-1 under normoxia (21% O₂) or hypoxia (1% O₂), infected with HSV-1 for 8 h. **g** Immunoblotting of HIF1α protein in THP-1 under physiological oxygen pressures (21% O₂ (≈160 mmHg O₂), 2.7% O₂ (≈20 mmHg O₂) and 1% O₂ (≈7 mmHg O₂)). **h** qPCR of *IFN-β* and *CXCL10* mRNA in THP-1 under physiological oxygen pressure (21% O₂, 2.7% O₂ and 1% O₂), infected without or with VSV (left two panels) or HSV-1 (right two panels). Data in (**a–e**, **h**) are presented as mean ± S.D., two-way ANOVA; *n* = 3 biological independent experiments. Data in (**f**) are presented as mean ± S.D., two-tailed Student's *t* test; *n* = 5 biological independent samples. Data in (**g**) are representative from three independent experiments. See also Supplementary Fig. 1. Source data are provided as a Source data file.

suggesting that *EGLN1* regulates antiviral immune responses independently of *VHL*.

To determine whether the enzymatic activity of EGLN1 is required for the positive regulation of cellular antiviral immune responses, we examined the activity of a catalytically inactivated EGLN1 mutant, in which histidine 313 was mutated to alanine (H313A)[59]. In HEK293T cells, SeV-induced expression of *IFN-β* was enhanced by ectopic expression of wild-type *EGLN1* (WT), but not of *EGLN1* mutant (H313A) (Supplementary Fig. 3a and Fig. 2f). In *HIF1β⁻/⁻* H1299 cells, overexpression of wildtype EGLN1 (*EGLN1*-WT), but not of EGLN1-H313A by lentivirus infection, enhanced VSV-induced *IFN-β* expression (Supplementary Fig. 3b, c). Conversely, overexpression of *EGLN1*-WT, but not of EGLN1-H313A inhibited VSV mRNA expression (Supplementary Fig. 3c, right panel). Notably, ectopic expression of either *EGLN2* or *EGLN3* had no apparent enhancement on the induction of *IFN-β* and *CXCL10* expression in H1299 cells and did not alter the number of VSV-GFP positive cells, suggesting that only *EGLN1* of the EGLN prolyl hydroxylase family has an obvious impact on the regulation of antiviral immune responses (Supplementary Fig. 3d–f).

We also used a specific inhibitor of EGLN prolyl hydroxylase, FG4592 (Roxadustat), for assays[60]. We confirmed that FG4592 did not affect H1299 apoptosis and viability when used in the range of 20 to 100 μM (Supplementary Fig. 4a). Treatment with FG4592 (20 μM) could induce the expression of *BNIP3* and *PGK1* (Supplementary Fig. 4b). Upon challenge with SeV, the expression of *IFN-β* and *CCL5* was suppressed by treatment with FG4592 in a dose-dependent manner (Supplementary Fig. 4c). Furthermore, in response to HSV-1 infection, or poly (I:C), or poly (dA:dT) transfection, treatment with FG4592 markedly inhibited *IFN-β* expression (Fig. 2g). Consistent with these findings, treatment with FG4592 promoted HSV-1-GFP virus replication in H1299 cells (Fig. 2h and Supplementary Fig. 4d). Similar results were obtained in THP-1 cells (Supplementary Fig. 4e–h).

Collectively, these data suggest that *EGLN1* is a key mediator of oxygen-mediated potentiation of antiviral innate immunity and that enzymatic activity is required for EGLN1 to positively regulate antiviral immune responses.

## Disruption of *Egln1* in mice results in increased susceptibility to lethal viral infection

In mouse embryonic fibroblast cells, we found that VSV- and HSV-induced the expression of *Ifn-β* was also largely lower in *Egln1⁻/⁻* MEF cells than in *Egln1⁺/⁺* MEF cells, and HSV-1-GFP-positive cells were more abundant in *Egln1⁻/⁻* MEF cells than in *Egln1⁺/⁺* MEF cells (Supplementary Fig. 5a, b), suggesting an evolutionarily conserved function of *Egln1* in antiviral immune responses. To further determine whether *Egln1* specifically modulates antiviral responses in response to viral infection, rather than having a broad transcriptional role, we performed genome-wide RNA sequencing (RNA-seq) analysis using *Egln1⁺/⁺* and *Egln1⁻/⁻* MEF cells infected without or with VSV. VSV-induced antiviral genes were apparently reduced in *Egln1⁻/⁻* MEF cells compared to those in *Egln1⁺/⁺* MEF cells (Supplementary Fig. 5c). Furthermore, gene ontology (GO) enrichment analysis revealed that the reduced genes resulting from *Egln1* loss were enriched in the category of innate immune response genes and response to virus genes (Supplementary Fig. 5d). Kyoto Encyclopedia of Genes and Genomes (KEGG) pathway enrichment analysis showed that these genes were enriched in the JAK-STAT and RLR signaling pathways (Supplementary Fig. 5e). Thus, upon viral challenge, *Egln1* mainly affected the expression of antiviral response genes, further strengthening the specific role of *Egln1* in modulating antiviral responses.

We therefore turned to a mouse model to study the function of *Egln1* in vivo. Since *Egln1*-knockout mice are embryonic lethal[61], we generated *Egln1* conditional knockout mice (Fig. 3a). Cre recombinase-mediated deletion resulted in the loss of exons 2 and 3 of *Egln1* in *Egln1*ᶠˡ/ᶠˡ mice (Fig. 3a). We obtained bone marrow-derived dendritic cells (BMDCs) from Cre-ER *Egln1⁺/⁺* and Cre-ER *Egln1*ᶠˡ/ᶠˡ mice and added 4OH-tamoxifen to culture medium to disrupt *Egln1* in Cre-ER *Egln1*ᶠˡ/ᶠˡ BMDCs and challenge with virus (Fig. 3b). In response to VSV or HSV-1 infection, the expression of *Ifn-β, Cxcl10, Ccl5* or *Ifit1* was largely reduced in Cre-ER *Egln1*ᶠˡ/ᶠˡ BMDC cells compared with that in Cre-ER *Egln1⁺/⁺* BMDC cells (Supplementary Fig. 6a, b). Consistently, upon challenge with VSV-GFP virus, virus replication was higher in Cre-ER *Egln1*ᶠˡ/ᶠˡ BMDC cells than in Cre-ER *Egln1⁺/⁺* BMDC cells (Fig. 3c, d and Supplementary Fig. 6c). In addition, Ifn-β levels were lower in Cre-ER *Egln1*ᶠˡ/ᶠˡ BMDC cells than in Cre-ER *Egln1⁺/⁺* BMDC cells (Fig. 3e).

Intraperitoneal injection of tamoxifen led to efficient deletion of *Egln1* in mouse organs, such as spleen, liver, and lung (Fig. 3f, g). Thus, the strategy to knock out *Egln1* in mice was reliable and successful, and the deletion of *Egln1* exhibits no obvious effects on the expression of *Ifn-β* or *Isg15* in the lung, spleen, and liver without virus infection (Supplementary Fig. 6d–f). After intraperitoneal injection of VSV, much more severe lung injury in Cre-ER *Egln1*ᶠˡ/ᶠˡ mice than Cre-ER *Egln1⁺/⁺* mice was observed as revealed by Hematoxylin and eosin (H & E) staining (Fig. 3h). Consistent with these findings, there was more Ifn-β in the serum of Cre-ER *Egln1⁺/⁺* mice than Cre-ER *Egln1*ᶠˡ/ᶠˡ mice (Fig. 3i), and Cre-ER *Egln1*ᶠˡ/ᶠˡ mice were more susceptible to VSV infection than Cre-ER *Egln1⁺/⁺* mice (Fig. 3j). Consistent with this, the expression of *Ifn-β, Cxcl10, Ccl5* and *Ifit1* was higher in the lung of Cre-ER *Egln1⁺/⁺* mice than Cre-ER *Egln1*ᶠˡ/ᶠˡ mice (Supplementary Fig. 6g), while the VSV mRNA levels were much lower in the lung of Cre-ER *Egln1⁺/⁺* mice than Cre-ER *Egln1*ᶠˡ/ᶠˡ mice (Supplementary Fig. 6h).

We then blocked the activity of EGLN in mice by oral gavage of FG4592 (10 mg/kg) (Fig. 3k). Treatment of mice with FG4592 significantly inhibited the expression of *Ifn-β, Cxcl10* and *Ifit1* in the spleen and liver of mice challenged with VSV compared with treatment with vehicle control treatment (Supplementary Fig. 6i, j). FG4592-treated mice were more susceptible to VSV infection than vehicle-treated mice (Fig. 3l).

These data suggest that *Egln1* promotes antiviral immunity in vivo.

## Disruption of *egln1* in zebrafish results in increased susceptibility to lethal viral infection

EGLN1 is evolutionarily conserved and zebrafish have two *egln1* genes, *egln1a* and *egln1b* (Supplementary Fig. 7a). We have previously

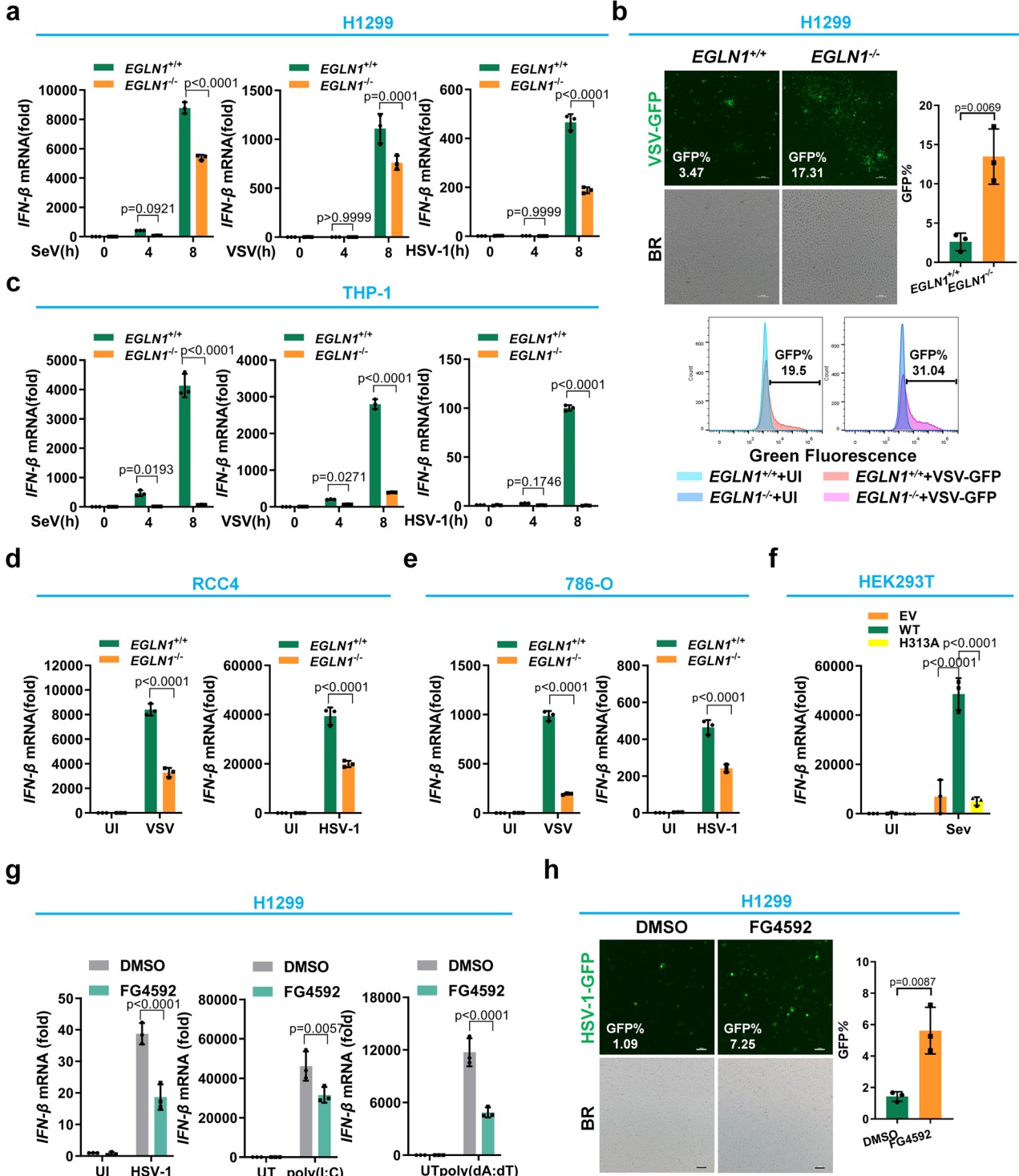

**Fig. 2 | *EGLN1* positively regulates cellular antiviral immune responses depending on its enzymatic activity. a** qPCR analysis of *IFN-β* mRNA in *EGLN1*⁺ᐟ⁺ and *EGLN1*⁻ᐟ⁻ H1299 cells infected with SeV, VSV or HSV-1 for 0, 4 and 8 h. **b** *EGLN1*⁺ᐟ⁺ and *EGLN1*⁻ᐟ⁻ H1299 cells were infected without (UI) or with VSV-GFP virus for 12 h, and viral infectivity was detected by fluorescence microscopy (top panels) or flow cytometry analysis (bottom panels). **c** qPCR analysis of *IFN-β* mRNA in *EGLN1*⁺ᐟ⁺ and *EGLN1*⁻ᐟ⁻ THP-1 cells infected with SeV, VSV or HSV-1 for 0, 4 or 8 h. **d** qPCR analysis of *IFN-β* mRNA in *EGLN1*⁺ᐟ⁺ and *EGLN1*⁻ᐟ⁻ RCC4 cells infected without (UI) or with VSV or HSV-1 for 8 h. **e** qPCR analysis of *IFN-β* mRNA in *EGLN1*⁺ᐟ⁺ and *EGLN1*⁻ᐟ⁻ 786-O cells infected without (UI) or with VSV or HSV-1 for 8 h. **f** qPCR analysis of *IFN-β* mRNA in HEK293T cells transfected with the HA empty vector (EV), the plasmid expressing HA-EGLN1 (WT) or the plasmid expressing the enzymatically inactive mutant of

EGLN1 (H313A) for 24 h, followed by infection without (UI) or with SeV for 8 h. **g** qPCR analysis of *IFN-β* in H1299 cells treated with DMSO (vehicle control) or FG4592 (20 μM) for 6 h, followed by infected without (UI) or with HSV-1 for 8 h, or transfected without (UT) or with poly (I:C) for 8 h, or transfected without (UT) or with poly (dA:dT) for 8 h. **h** H1299 cells were treated with DMSO (vehicle control) or FG4592 (20 μM) for 6 h, followed by infection without (UI) or with VSV-GFP virus for 12 h, and viral infectivity was detected by fluorescence microscopy analysis. BR, bright field. *EGLN1*⁻ᐟ⁻ H1299 cells are clonal. Data in (**a**, **c**–**g**) are presented as mean ± S.D., two-way ANOVA; *n* = 3 biological independent experiments. Data in (**b**, **h**) are presented as mean ± S.D., two-tailed Student's *t* test; *n* = 3 biological independent experiments. See also Supplementary Figs. 2–5. Source data are provided as a Source data file.

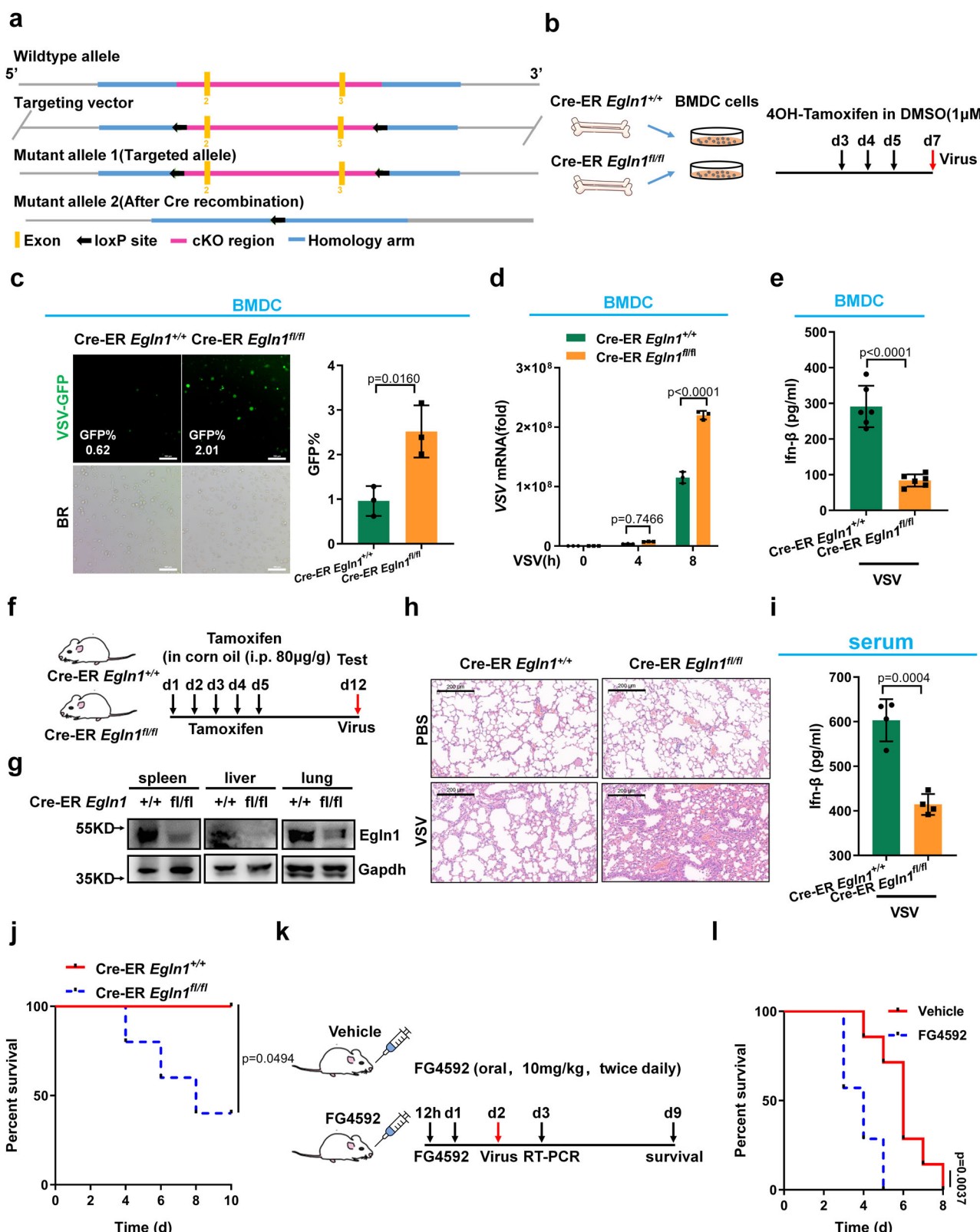

generated *egln1a⁻/⁻egln1b⁻/⁻* zebrafish by CRISPR/Cas9-mediated genome editing (Fig. 4a and Supplementary Fig. 7b, c)[62]. To further validate the role of *Egln1* in vivo, we took advantage of this zebrafish model. Overall, the homozygous (*egln1a⁻/⁻egln1b⁻/⁻*) zebrafish were physiologically indistinguishable from their wild-type siblings (Fig. 4b, left two panels). However, when challenged with grass carp reovirus (GCRV), a double-stranded RNA virus, *egln1a⁻/⁻egln1b⁻/⁻* zebrafish

exhibited a higher mortality rate compared to their wild-type (WT) siblings (*egln1a⁺/⁺egln1b⁺/⁺*) (Fig. 4b, c), and the expression of *mxc*, a well-defined antiviral-responsive genes in zebrafish, was much lower in GCRV-infected *egln1a⁻/⁻egln1b⁻/⁻* zebrafish compared to *egln1a⁺/⁺egln1b⁺/⁺* zebrafish (Supplementary Fig. 7d). Similarly, when challenged with spring viraemia of carp virus (SVCV), a single-stranded RNA virus, *egln1a⁻/⁻egln1b⁻/⁻* zebrafish exhibited a higher mortality rate

**Fig. 3 | Disruption of *Egln1* in mice results in increased susceptibility to lethal viral infection. a** Scheme for CRISPR/Cas9-mediated genome editing. **b** Scheme for obtaining wild-type and *Egln1*-deficient BMDC (Cre-ER (ER= estrogen receptor) *Egln1*⁺/⁺ and Cre-ER *Egln1*^fl/fl) and viral infection. **c** Cre-ER *Egln1*⁺/⁺ and Cre-ER *Egln1*^fl/fl BMDC were infected with VSV-GFP for 12 h (*n* = 3 biological independent experiments). **d** qPCR analysis of VSV mRNA in Cre-ER *Egln1*⁺/⁺ and Cre-ER *Egln1*^fl/fl BMDC infected with VSV for 0, 4, and 8 h. **e** ELISA assay of Ifn-β in supernatants of Cre-ER *Egln1*⁺/⁺ and Cre-ER *Egln1*^fl/fl BMDC infected with VSV for 12 h (*n* = 6 biological independent samples). **f** Scheme for generation of Cre-ER *Egln1*⁺/⁺ and Cre-ER *Egln1*^fl/fl mice and viral infection. **g** Validation of Egln1 protein in indicated tissues of Cre-ER *Egln1*⁺/⁺ and Cre-ER *Egln1*^fl/fl mice. **h** H & E-stained images of lung sections from tamoxifen-treated Cre-ER *Egln1*⁺/⁺ and Cre-ER *Egln1*^fl/fl mice injected intraperitoneally with PBS (phosphate buffer saline) or VSV (1 × 10⁷ plaque-forming units (PFU) per mouse) for 24 h. Scale bar = 200 μm. **i** ELISA assay of Ifn-β in serum from tamoxifen-treated Cre-ER *Egln1*⁺/⁺ and Cre-ER *Egln1*^fl/fl mice infected with VSV (1 × 10⁷ PFU per mouse) for 24 h (*n* = 4 biological independent samples). **j** Survival of Cre-ER *Egln1*⁺/⁺ and Cre-ER *Egln1*^fl/fl mice by injected intraperitoneally with VSV (1 × 10⁷ PFU per mouse) and monitored for 10 d. Statistical analysis was performed using the log-rank test (*n* = 5 for each group). **k** Scheme for FG4592 treatment (10 mg/kg) and viral infection. **l** Survival of mice treated with vehicle control or FG4592 twice daily for 2 days, followed by intraperitoneal injection of VSV (5 × 10⁷ PFU per mouse) and monitored for 8 days. Statistical analysis was performed using the log-rank test (*n* = 7 for each group). Data in (**c, e, i**) are presented as mean ± S.D., two-tailed Student's *t* test. Data in (**d**) are presented as mean ± S.D., two-way ANOVA; *n* = 3 biological independent experiments. Data in (**g, h**) are representative from three independent experiments. See also Supplementary Fig. 6. Source data are provided as a Source data file.

compared to their wild-type (WT) siblings (*egln1a*⁺/⁺*egln1b*⁺/⁺) (Supplementary Fig. 7e, f). Further assays revealed that the expression of *ifn1*, *pkz* and *mxc*, three well-defined antiviral-responsive genes in zebrafish, was much lower in both liver and kidney of SVCV-infected *egln1a*⁻/⁻*egln1b*⁻/⁻ zebrafish compared to *egln1a*⁺/⁺*egln1b*⁺/⁺ zebrafish (Supplementary Fig. 7g).

Next, we used FG4592 to block the activity of Egln in zebrafish[60]. First, we tested the toxicity of FG4592 on zebrafish larvae at different concentrations and then selected 10 μM for further assays (Supplementary Fig. 8a, b). Treatment of zebrafish larvae with FG4592 resulted in increased susceptibility to SVCV infection (Fig. 4d, e). Increased *vegf* expression in FG4592-treated zebrafish larvae indicated that FG4592 efficiently blocked the enzymatic activity of prolyl hydroxylases (Supplementary Fig. 8c). Consistent with these findings, the expression of *ifn1*, *mxc*, *lta* and *pkz* was greatly reduced in FG4592 treated zebrafish larvae upon challenge with SVCV compared to vehicle control-treated zebrafish larvae (Supplementary Fig. 8d). Furthermore, using the transgenic zebrafish line, Tg (ifnφ1: mCherry) (in which the fluorescent protein, mCherry, is driven by the zebrafish ifnφ1 promoter), we observed that treatment of zebrafish larvae with FG4592 significantly reduced fluorescent signaling upon challenge with SVCV compared with that in vehicle (DMSO) control-treated zebrafish larvae (Fig. 4f, g). To confirm the specificity of the effect of FG4592 on SVCV-induced IFN activation in zebrafish, we subsequently used another transgenic zebrafish line, Tg (lyz: DsRed2) (in which the fluorescent protein, DsRed2, is driven by the zebrafish lyz promoter). This fish line can be used to monitor myeloid precursors expressing lysozyme c gene (lyz) as well as neutrophils in the blood. Of note, compared to the control (DMSO), treatment with FG4592 had no apparent effect on the changes in red fluorescence signaling in SVCV-infected zebrafish larvae (Supplementary Fig. 8e, f). These data further validated that FG4592 may specifically affect SVCV-induced IFN activation.

Taken together, these data further support that *egln1a/b* facilitates antiviral immunity in vivo.

## EGLN1 interacts with IRF3 to enhance IRF3 activation

The enhancement exerted by *EGLN1* of innate antiviral responses against both RNA viruses (SeV and VSV) and DNA viruses (HSV-1) led us to hypothesize that *EGLN1* may act by regulating common factors downstream of the RLR and the cGAS-STING signaling pathways[28]. IRF3 was a potential target because it is not only a common effector of these two pathways, but it is also a key transcription factor for the induction of type I interferons (IFNs) and subsequent interferon-stimulated genes. We therefore investigated whether EGLN1 affects the function of IRF3 through protein-protein interactions. Colocalization assay showed that ectopically expressed IRF3 colocalized with ectopically expressed EGLN1 in the cytosol (Supplementary Fig. 9a). Co-immunoprecipitation assays showed that ectopically expressed *IRF3* pulled down ectopically expressed *EGLN1* in HEK293T cells and vice versa (Fig. 5a, b). In H1299 cells, endogenous IRF3 was co-

immunoprecipitated with endogenous EGLN1 in *IRF3*-intact (*IRF3*⁺/⁺) H1299 cells, whereas no endogenous immunoprecipitation between IRF3 and EGLN1 was detected in *IRF3*-null (*IRF3*⁻/⁻) H1299 cells (Fig. 5c). GST-tagged IRF3 expressed in *Escherichia coli* interacted with *E. coli*-expressed His-tagged *EGLN1* in vitro (Supplementary Fig. 9b), suggesting that EGLN1 directly associates with IRF3. The spatial colocalization and interaction between EGLN1 and IRF3 was further confirmed by in situ proximity ligation assay (PLA) (Fig. 5d). As shown in Fig. 5d, no red spots were detected when H1299 cells were transfected with the empty HA vector control, whereas transfection of HA-EGLN1 resulted in a large increase of red spots. Furthermore, ectopically expressed IRF3 also bound to ectopically expressed EGLN1 under hypoxia or by the treatment with FG4592 (Supplementary Fig. 9c, d). Domain mapping revealed that the C-terminal PH domain of EGLN1 is required for the interaction with IRF3, whereas the IAD domain of IRF3 is required for the interaction with EGLN1 (Fig. 5e–h).

Co-immunoprecipitation assays showed that ectopically expressed EGLN1, but not EGLN2 or EGLN3, could pull down ectopically expressed IRF3 in HEK293T cells (Supplementary Fig. 9e).Co-expression of EGLN1 with IRF3 could enhance the activity of IRF3 on the induction of *IFN-β* expression, but co-expression of either *EGLN2* or *EGLN3* did not affect the activity of IRF3 on the induction of *IFN-β* and *IFIT1* expression (Supplementary Fig. 10a, b), further supporting that only *EGLN1* of the EGLN prolyl hydroxylase family positively regulates IRF3 activation.

In *IRF3*-deficient H1299 cells (*IRF3*⁻/⁻), ectopic expression of *EGLN1* had no effect on either VSV- or HSV-1-induced *IFN-β* expression (Supplementary Fig. 10c). Furthermore, the replication of both VSV-GFP and HSV-1-GFP viruses was quite similar between the HA empty vector-transfected and HA-*EGLN1*-transfected *IRF3*⁻/⁻ H1299 cells (Supplementary Fig. 10d–g). These data suggest that IRF3 mediates the role of EGLN1 in antiviral immunity.

To further validate that type I IFN is an essential effector downstream of *EGLN1*, we used the Vero cell line, which is deficient for type I IFN production upon virus infection[63]. Ectopic expression of *EGLN1* had no apparent effect on VSV-GFP virus replication (Supplementary Fig. 10h, i), suggesting that type I IFN is indispensable for mediating the role of *EGLN1* in the regulation of antiviral immune responses.

Taken together, these data suggest that *EGLN1* exerts its function in antiviral immunity specifically by modifying IRF3 activity.

## EGLN1 hydroxylates the proline 10 of IRF3

Using mass spectrometry (MS) analysis, we identified that IRF3 can be hydroxylated by EGLN1 at proline 10 (P10) (Fig. 6a), which appears to be evolutionarily conserved (Fig. 6b). A commercial anti-hydroxyproline antibody is available (Abcam; Cat# ab37067), which has recently been successfully used to identify proline hydroxylation in DYRK1 kinases[64]. This antibody was readily able to pull-down wild-type IRF3 (WT) ectopically expressed in HEK293T cells, but not IRF3 mutant (in which the proline 10 was mutated to alanine) (P10A)

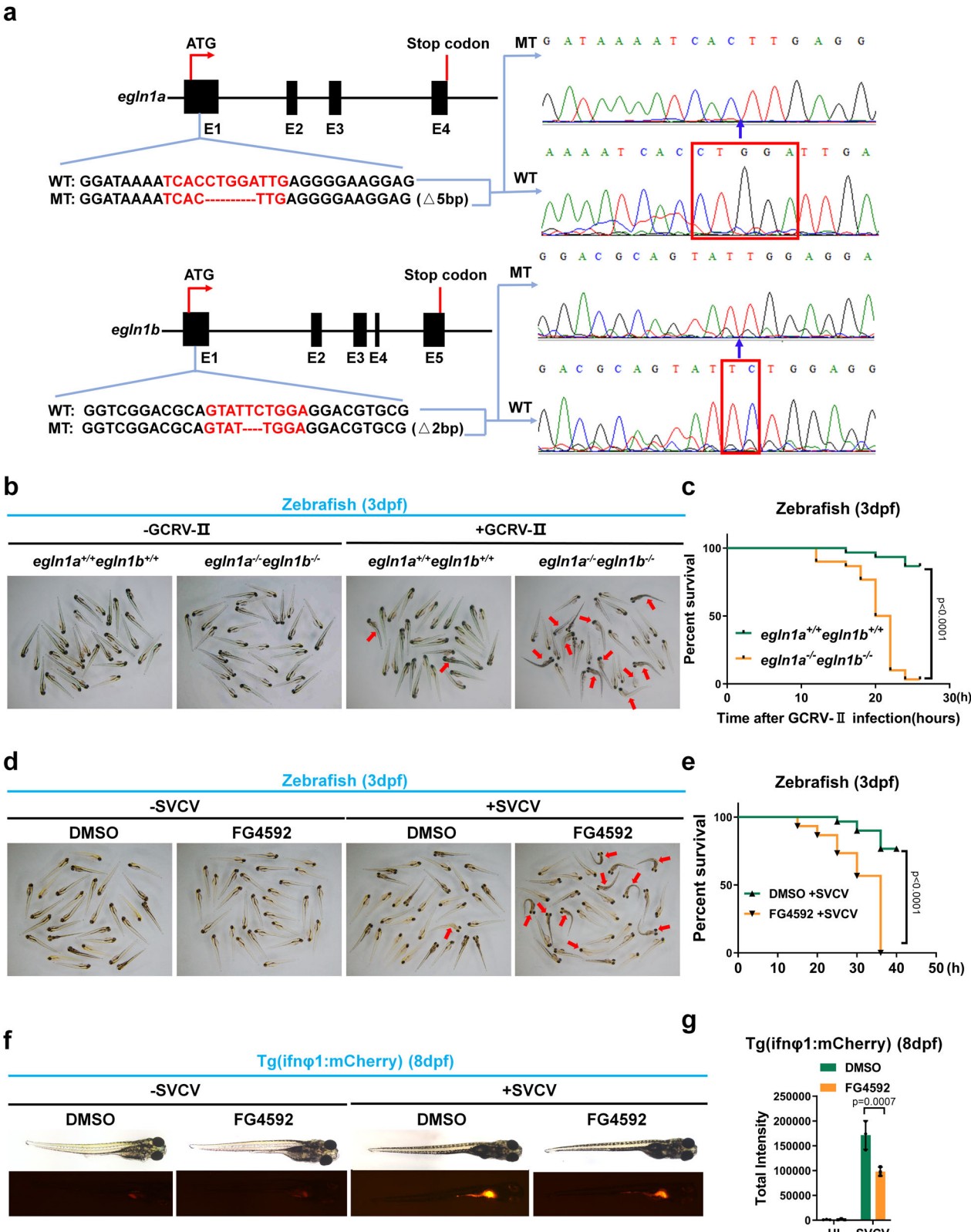

(Supplementary Fig. 11a), indicating that IRF3 might be hydroxylated at the proline 10. To further confirm this hydroxylated site in IRF3, we developed a specific antibody against the hydroxylated form of IRF3 proline 10 (anti-IRF3-P10-OH antibody); its specificity was confirmed by dot blot assays (Supplementary Fig. 11b). When WT (Flag-IRF3) or mutant IRF3 (Flag-IRF3-P10A) was ectopically expressed in HEK293T cells, only WT IRF3 could be detected by the anti-IRF3-P10-

OH antibody in the total cell lysate and in the precipitate obtained with anti-Flag conjugated agarose beads (Fig. 6c), confirming the specificity of the antibody. Furthermore, the prolyl hydroxylation was only detected in *IRF3*-intact H1299 cells (*IRF3*[+/+]), but not in *IRF3*-null H1299 cells (*IRF3*[−/−]) (Fig. 6d), again validating the specificity of the antibody.

We then tested whether EGLN1 could catalyze the hydroxylation of IRF3 at proline 10 using the anti-IRF3-P10-OH antibody. Ectopic

**Fig. 4 | Disruption or inhibition of *egln1* in zebrafish results in increased susceptibility to lethal viral infection. a** Scheme for CRISPR/Cas9-mediated genome editing of the *egln1a* and *egln1b* gene locus; and the resulting sequence information in *egln1a* or *egln1b* null zebrafish. **b** Representative images of wild-type (*egln1a⁺/⁺egln1b⁺/⁺*) and *egln1a* and *egln1b* double knockout (*egln1a⁻/⁻egln1b⁻/⁻*) zebrafish larvae (3 dpf, *n* = 30) infected without (−) or with (+) GCRV-II for 18 h. The dead larvae showed lack of movement, curved body, and a bodily degeneration as indicated by the red arrows. **c** Survival (Kaplan−Meier curve) of wild-type (*egln1a⁺/⁺egln1b⁺/⁺*) and *egln1a* and *egln1b* double knockout (*egln1a⁻/⁻egln1b⁻/⁻*) zebrafish larvae (3 dpf) infected without (−) or with (+) GCRV-II for and monitored for 30 h. Statistical analysis was performed using the log-rank test (*n* = 30 for each group). **d** Representative images of zebrafish larvae (3 dpf) treated with DMSO (vehicle control) or FG4592 (10 μM) for 24 h, followed by infection without (−) or with SVCV

($-6 \times 10^7$ TCID$_{50}$/ml) for 24 h. The dead larvae showed lack of movement, curved body, and a bodily degeneration as indicated by the red arrows. **e** Survival (Kaplan−Meier curve) of zebrafish larvae (3 dpf) treated with DMSO (vehicle control) or FG4592 (10 μM) for 24 h, followed by infection without (−) or with SVCV ($-6 \times 10^7$ TCID$_{50}$/ml) and monitored for 40 h. Statistical analysis was performed using the log-rank test (*n* = 30 for each group). **f** Representative images of Tg(ifnφ1: mCherry) zebrafish larvae (8 dpf) treated with DMSO (vehicle control) or FG4592 (10 μM) for 12 h, followed by infected without (−) or with SVCV ($-6 \times 10^7$ TCID$_{50}$/ml) for 24 h. **g** Quantitation of total intensity in (**f**). UI uninfected. Data in (**b**, **d**, **f**) are representative from three independent experiments. Data in (**g**) are presented as mean ± S.D., two-way ANOVA; *n* = 3 biological independent experiments. See also Supplementary Figs. 7 and 8. Source data are provided as a Source data file.

expression of WT EGLN1 catalyzed prolyl hydroxylation of ectopically expressed IRF3, but ectopic expression of the catalytically inactivated EGLN1 mutant (H313A) did not, like the empty vector control (EV) control in HEK293T cells (Supplementary Fig. 11c) and in *EGLN1⁻/⁻* H1299 cells (Fig. 6e). When the IRF3-P10A mutant was co-expressed with wild-type EGLN1 or mutant (H313A), no band could be detected by the anti-IRF3-P10-OH antibody, supporting that the EGLN1 catalyzes IRF3 proline-10 hydroxylation (Supplementary Fig. 11c and Fig. 6e). Depletion of endogenous *EGLN1* in H1299 cells resulted in a complete loss of IRF3 proline 10 hydroxylation (Fig. 6f).

As expected, treatment under hypoxia or addition of DMOG (a hydroxylase inhibitor), largely reduced IRF3 prolyl hydroxylation (Fig. 6g, h). In response to VSV infection, IRF3 prolyl hydroxylation initially increased (2 h) and then decreased (4 h) (Fig. 6i, j), suggesting a transient physiological role for IRF3 prolyl hydroxylation in response to viral infection. Notably, the interaction between IRF3 and EGLN1 was stable during viral infection (Supplementary Fig. 11d). To further understand the reason, we then examined whether the hydroxylase activity of EGLN1 was altered in the process of viral infection by using HIF1α hydroxylation as an indicator. As shown in Supplementary Fig. 11e, f, the hydroxylase activity of EGLN1 was initially increased upon VSV infection for 2 h, and subsequently decreased upon VSV infection for 4 h (Supplementary Fig. 11e, f). It appeared that the dynamics of IRF3 hydroxylation during viral infection was due to changes in the hydroxylase activity of EGLN1 during viral infection.

To determine whether IRF3 is a bona fide substrate of EGLN1, we followed a previously reported in vitro hydroxylation assay protocol[65]. To verify that this protocol worked correctly in our hands, we established a positive control by using the ODD domain of HIF1α (400-575aa) for the assays[65]. We incubated bacterially expressed HIF1α-ODD together with bacterially expressed GST-EGLN1 protein or bacterially expressed GST protein (control) in the reaction containing essential cofactors required for the enzymatic activity of EGLN prolyl hydroxylases[65]. Immunoblotting by using a commercially available anti-hydroxylated HIF-1α antibody confirmed that the ODD domain of HIF-1α (400−575aa) was sufficiently hydroxylated by GST-EGLN1 (Fig. 6k). Further mass spectrometry analysis showed that P564 in the HIFα-ODD was easily identified as hydroxylated in the presence of GST-EGLN1, but not in the presence of GST protein (Supplementary Fig. 11g). We then used this protocol to perform an in vitro hydroxylation assay for IRF3. As shown in Fig. 6l, IRF3 was hydroxylated by GST-EGLN1 in the presence of both Fe$^{2+}$ and α-KG, but not in the absence of either Fe$^{2+}$ or α-KG. Fe$^{2+}$ or α-KG are the two essential cofactors required for EGLN prolyl hydroxylase activity[7], Mass spectrometry analysis showed that P10 in IRF3 was also readily identified as being hydroxylated in the presence of GST-EGLN1, but not in the presence of the GST protein control (Fig. 6m).

Taken together, these data suggest that IRF3 is a bona fide substrate of EGLN1 that can be sufficiently hydroxylated by EGLN1.

## EGLN1 promotes IRF3 phosphorylation, dimerization, and nuclear translocation

To address the biological consequences of EGLN1-catalyzed IRF3 prolyl hydroxylation, we first investigated whether *EGLN1* influences IRF3 protein stability. Ectopic expression of an increasing amounts of either wild-type EGLN1 or H313A (catalytically inactivated EGLN1) did not affect the protein level of co-transfected wild-type IRF3 (Flag-IRF3-WT) (Fig. 7a). IRF3 protein levels were similar between *EGLN1⁺/⁺* and *EGLN1⁻/⁻* H1299 cells (Fig. 7b). Treatment with increasing amounts of FG4592 did not alter IRF3 protein levels in H1299 cells (Supplementary Fig. 12a). In addition, treatment with hypoxia did not alter IRF3 protein levels in H1299 cells (Fig. 7c). Thus, EGLN1 does not affect the protein stability of IRF3.

We then examined IRF3 phosphorylation, dimerization and nuclear translocation, three key events that determine IRF3 activation[29], during viral infection and in the presence or absence of *EGLN1*. Remarkably, in response to SeV, VSV or HSV-1 infection, IRF3 phosphorylation and dimerization were attenuated in *EGLN1⁻/⁻* H1299 cells or *EGLN1⁻/⁻* THP-1 cells compared to *EGLN1⁺/⁺* H1299 cells or *EGLN1⁺/⁺* THP-1 cells (Supplementary Fig. 12b and Fig. 7d, e). Similar phenomena were observed when comparing *Egln1⁺/⁺* and *Egln1⁻/⁻* MEF cells (Supplementary Fig. 12c). Consistently, in *EGLN1⁻/⁻* H1299 cells, ectopic expression of wild-type *EGLN1* promoted IRF3 phosphorylation and dimerization upon VSV infection, whereas ectopic expression of H313A *EGLN1* did not, like the empty vector (EV) control (Fig. 7f). Furthermore, treatment with FG4592 also impaired VSV- and HSV-1-induced IRF3 phosphorylation and dimerization in H1299 cells (Supplementary Fig. 12d, e).

Upon SeV infection, more IRF3 was translocated into the nuclei of *EGLN1⁺/⁺* HEK293T cells compared to *EGLN1⁻/⁻* HEK293T cells (Fig. 7g). Furthermore, reconstitution of wild-type *EGLN1* increased IRF3 nuclear translocation in response to SeV infection, whereas H313A *EGLN1* did not, like empty vector (EV) control (Fig. 7h).

These data suggest that EGLN1 promotes IRF3 phosphorylation, dimerization, and nuclear translocation depending on its prolyl hydroxylase activity, thereby enhancing IRF3 activation.

## Hydroxylation of IRF3 at proline 10 enhances IRF3 activation in cellular antiviral immune responses

To further determine whether the enhancement of IRF3 activity by EGLN1 is mediated by EGLN1-catalyzed hydroxylation on IRF3 proline 10 (P10), we compared the transcriptional activity of wild-type IRF3 (WT) and P10A IRF3 using promoter assays. The luciferase activity of both an *IFN-β* luciferase reporter and an ISRE luciferase reporter driven by ectopic expression of P10A IRF3 was much lower than that driven by ectopic expression of wild-type IRF3 in HEK293T cells (Supplementary Fig. 13a and Fig. 8a). Similar results were obtained in *IRF3⁻/⁻* H1299 cells (Supplementary Fig. 13b and Fig. 8b) and *Irf3/irf7*-double deficient (*Irf3⁻/⁻Irf7⁻/⁻*) MEF cells (Supplementary Fig. 13c, d). In response to SeV, VSV, or HSV-1 infection, the upregulation of *IFN-β* mRNA was

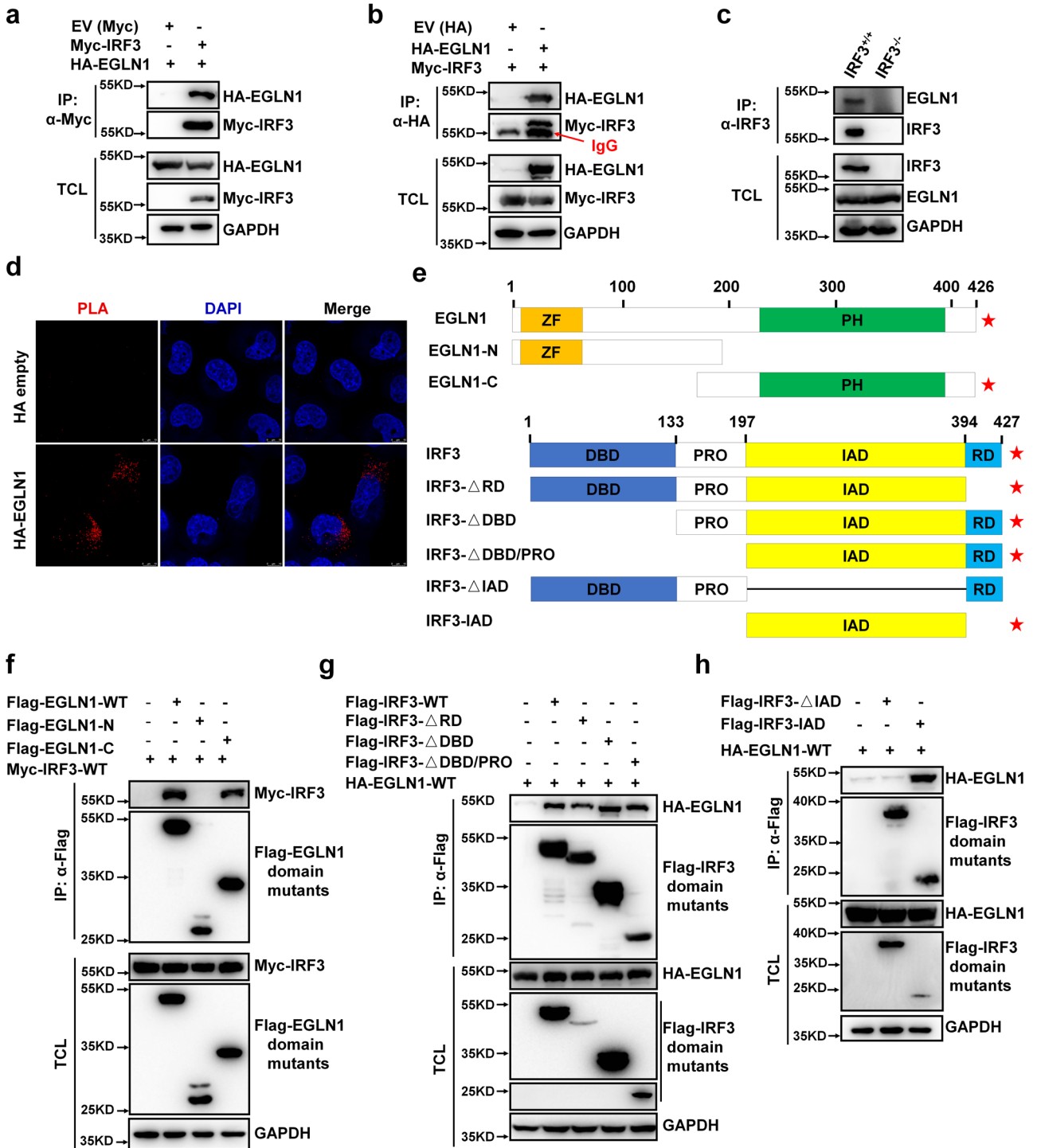

**Fig. 5 | EGLN1 interacts with IRF3. a, b** Co-immunoprecipitation of Myc-IRF3 with HA-EGLN1 and vice versa. HEK293T cells were co-transfected with the indicated plasmids for 24 h. Anti-Myc or anti-HA antibody conjugated agarose beads were used for immunoprecipitation and the interaction was detected by immunoblotting with the indicated antibodies. **c** Endogenous interaction between IRF3 and EGLN1 in *IRF3*-deficient (*IRF3*⁻/⁻) or wild-type H1299 cells (*IRF3*⁺/⁺). **d** In situ PLA assays of the EGLN1-IRF3 interaction in H1299 cells with the indicated combinations using anti-HA and anti-IRF3 antibodies, scale bar = 10 μm. **e** Schematic of EGLN1 domains interacting with IRF3 domains. The positive result of the interaction is indicated by the (★) signs. **f** Co-immunoprecipitation analysis of Myc-IRF3 with Flag-EGLN1 truncated mutants. HEK293T cells were co-transfected with the indicated plasmids. Anti-Flag antibody conjugated agarose beads were used for immunoprecipitation and the interaction was analyzed by immunoblotting with

anti-Myc antibody. Flag-EGLN1 fragments (WT: full length; N, 1–196 aa; C, 130–426 aa). **g, h** Co-immunoprecipitation analysis of HA-EGLN1 with Flag-IRF3 truncated mutants. HEK293T cells were co-transfected with the indicated plasmids. Anti-Flag antibody conjugated agarose beads were used for immunoprecipitation and the interaction was analyzed by immunoblotting with anti-HA antibody. Flag-IRF3 fragments (WT: full length; ΔRD, 1–394 aa; ΔDBD, 133–427 aa; ΔDBD&PRO, 197–427 aa; ΔIAD, 1–197 & 394–427 aa; IAD, 197–394 aa). Flag-IRF3-ΔDBD&PRO expression was relatively lower compared to other fragments, its band was independently excised for longer exposure. EV empty vector, IP immunoprecipitation, TCL total cell lysates, PLA proximity ligation assay. Data in (**a–d, f–h**) are representative from three independent experiments. See also Supplementary Figs. 9 and 10. Source data are provided as a Source data file.

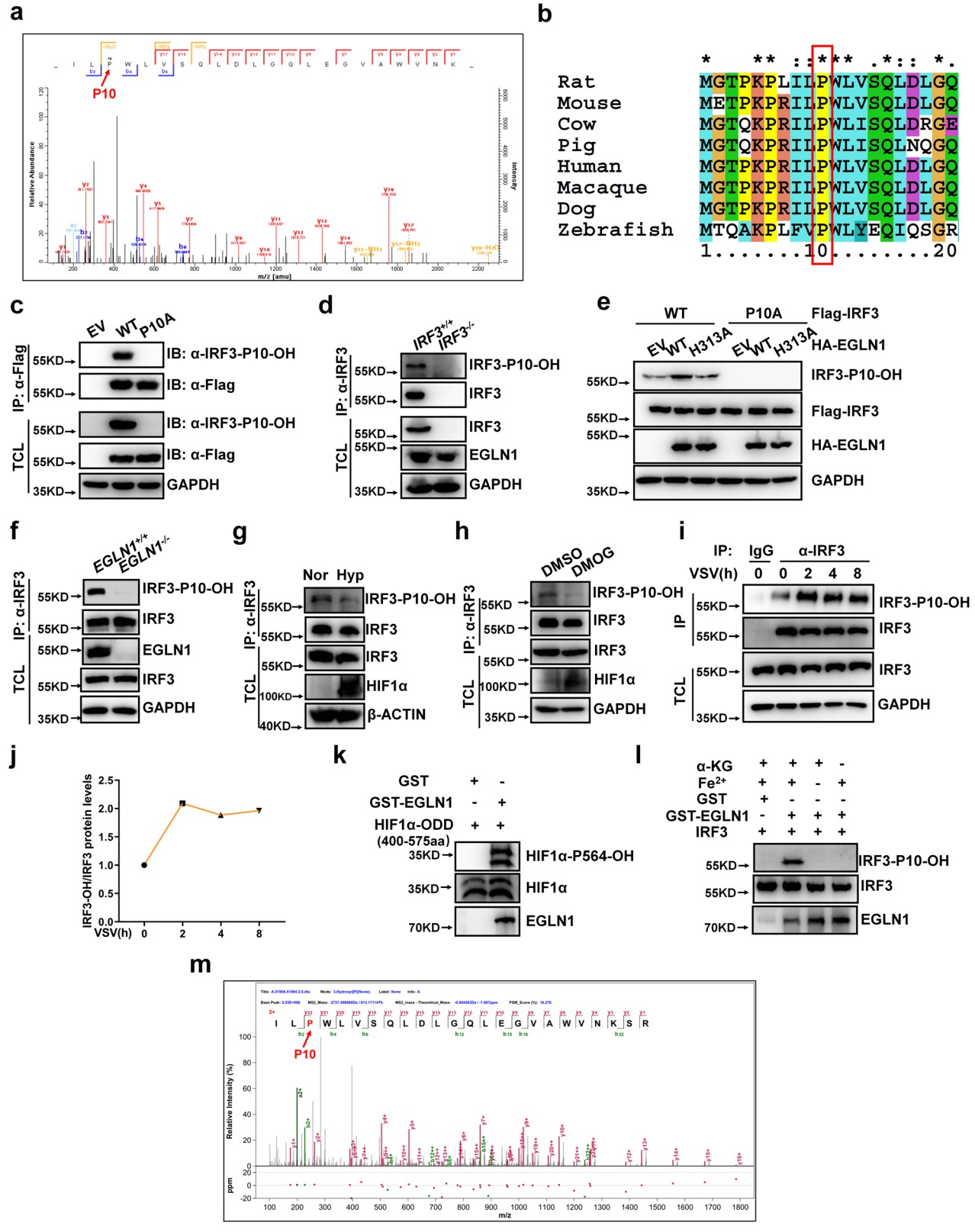

significantly lower in HEK293T cells or *IRF3⁻ᐟ⁻* H1299 cells expressing ectopic P10A IRF3 than in cells expressing ectopic wild-type IRF3 (Fig. 8c and Supplementary Fig. 13e). Ectopic expression of EGLN1 in *IRF3⁻ᐟ⁻* H1299 cells enhanced ectopic WT IRF3-mediated upregulation of *IFN-β, CXCL10* and *ISG15* mRNA, but had no effect on the effect of ectopic P10A IRF3 (Fig. 8d). Similar results were obtained in *Irf3⁻ᐟ⁻ Irf7⁻ᐟ⁻* MEF cells (Fig. 8e). Furthermore, disruption of *EGLN1* in H1299 cells

suppressed ectopic WT IRF3-induced upregulation of *IFN-β, CXCL10* and *ISG15* mRNA, but had no effect on the effect of ectopic P10A IRF3 (Supplementary Fig. 13f).

In addition, VSV-GFP virus infectivity monitored by fluorescence microscopy, immunoblotting and flow cytometry, was significantly reduced in *IRF3⁻ᐟ⁻* H1299 cells expressing ectopic WT IRF3, compared to the same cells expressing ectopic P10A IRF3 (Fig. 8f and

**Fig. 6 | EGLN1 hydroxylates IRF3 at proline 10. a** The hydroxylated residue in IRF3 was identified by mass spectrometry analysis. **b** Sequence alignment of partial IRF3 (1–20 amino acids). **c** HEK293T were transfected with the Flag empty vector (EV), Flag-IRF3 or Flag-IRF3-P10A, followed by immunoprecipitation and immunoblotting with the indicated antibodies. **d** Cell lysates from *IRF3*+/+ and *IRF3*−/− H1299 were extracted, followed by immunoprecipitation and immunoblotting with the indicated antibodies. **e** *EGLN1*−/− H1299 were transfected with Flag-IRF3-WT or Flag-IRF3-P10A, together with the HA empty vector (EV), HA-EGLN1-WT (WT), HA-EGLN1-H313A (H313A), followed by immunoprecipitation and immunoblotting with the indicated antibodies. **f** Cell lysates from *EGLN1*+/+ and *EGLN1*−/− H1299 were extracted, followed by immunoprecipitation and immunoblotting with the indicated antibodies. **g** Cell lysates from THP-1 under normoxia (21% $O_2$) or hypoxia (1% $O_2$) for 12 h were extracted, followed by immunoprecipitation and immunoblotting with the indicated antibodies. **h** Cell lysates from THP-1 treated with DMSO (vehicle control) or DMOG (1 mM) for 6 h were extracted, followed by immunoprecipitation and immunoblotting with the indicated antibodies. **i** Cell lysates from H1299

infected with VSV virus for the indicated time were extracted, followed by immunoprecipitation and immunoblotting with the indicated antibodies. **j** Quantitation of the intensity of hydroxylated IRF3/total IRF3 protein in (**i**). **k** Bacterially expressed HIF1α (400-575aa) (GST tag has been removed using thrombin) incubated with bacterially expressed GST or bacterially expressed GST-EGLN1 for hydroxylation reaction and then detected by immunoblotting with anti-HIF1α-P564-OH antibody. **l** In vitro hydroxylation assay to detect bacterially expressed IRF3 hydroxylated by bacterially expressed GST-EGLN1. **m** Bacterially expressed IRF3 (GST tag has been removed using thrombin) was incubated with bacterially expressed GST-EGLN1 for hydroxylation reaction and then identified by mass spectrometry analysis. Hydroxylated IRF3-P10 is indicated by the red arrow, and the percentage of IRF3 hydroxylation is 10.34%. EV empty vector, IP immunoprecipitation, TCL total cell lysates. Data in (**c**–**i**, **k**, **l**) are representative from three independent experiments. See also Supplementary Fig. 11. Source data are provided as a Source data file.

Supplementary Fig. 13g). In response to SeV infection, ectopic WT IRF3 was readily detectable in the nuclei of *IRF3*−/− H1299 cells, whereas IRF3-P10A was retained in the cytosol of *IRF3*−/− H1299 cells (Fig. 8g). In addition, upon VSV infection, ectopic WT IRF3 pulled down more endogenous TBK1 than ectopic P10A IRF3 did (Supplementary Fig. 13h), suggesting that IRF3 hydroxylation on IRF3 proline 10 promotes the interaction between IRF3 and TBK1.

Taken together, these data suggest that EGLN1 enhances antiviral innate immunity by specifically catalyzing the hydroxylation of IRF3 at proline 10.

## Irf3 prolyl hydroxylation deficiency attenuates antiviral innate immunity in mice

In mice, Irf3 prolyl hydroxylation at P10 appeared to be reduced after VSV infection for 24 h (Fig. 9a–d), suggesting that a negative feedback mechanism might exist by altering Irf3 hydroxylation in response to viral infection.

To investigate the role of Irf3 hydroxylation in vivo[66], we generated an Irf3_P10A (mouse homolog of human IRF3 P10A) mutant mouse model using CRISPR/Cas9-mediated genome editing (Supplementary Fig. 14a, b). Under standard housing conditions, Irf3 P10A mice appeared normal and fertile, and were indistinguishable from their wild-type siblings. As expected, Irf3 prolyl hydroxylation was undetectable at P10 in the spleen and lungs of Irf3_P10A mice (Fig. 9e, f). Flow cytometry analysis showed that the lymphocyte numbers or proportions in spleen and intestinal lymph nodes were similar between WT and Irf3_P10A mice (Supplementary Fig. 14c–e).

Compared to WT bone marrow-derived macrophages (BMDMs) and dendritic cells (BMDCs), the induction of *Ifn-β* and *Cxcl10* expression upon VSV challenge was much lower in Irf3_P10A BMDMs and BMDCs (Supplementary Fig. 15a and Fig. 9g), and IRF3 dimerization and phosphorylation were reduced in Irf3_P10A BMDCs compared to WT BMDCs upon VSV infection (Fig. 9h). Similar results were observed when WT and Irf3_P10A BMDMs or BMDCs were challenged with HSV-1 (Supplementary Fig. 15b and Fig. 9i, j). Consistent with these findings, BMDCs from WT mice produced more Ifn-β protein than Irf3_P10A mice upon VSV infection (Supplementary Fig. 15c).

We then challenged the mice by intraperitoneally (i.p.) injection of VSV or HSV-1 virus. *Irf3* P10A mice were more susceptible to VSV or HSV-1 challenge than WT mice (Fig. 9k, l). H & E staining showed more injury in the lungs of Irf3_P10A mice, compared to the lungs of WT mice after challenge with VSV or HSV-1 (Fig. 9m). The expression of *Ifn-β, Cxcl10* or *Ifit1* in spleen and lung was significantly lower in Irf3 P10A mice than in WT mice (Supplementary Fig. 15d–g).

To further validate that *EGLN1* mediates the phenotypic difference between WT and Irf3_P10A mice in response to viral infection, we used FG4592 (10 mg/kg) to block EGLN1 activity in WT and Irf3_P10A mice and then challenged them with VSV. VSV-induced *Ifn-β* and *Cxcl10*

expression in spleen and liver showed no difference between WT and Irf3_P10A mice after treatment with FG4592 (Supplementary Fig. 15h, i), suggesting that EGLN1 indeed mediates the difference between WT and Irf3_P10A mice in response to viral infection.

To determine whether HIF1α was involved in the phenotypic difference between WT and Irf3_P10A mice in response to viral infection, we used PX478, a specific HIF1α inhibitor[51,52], to block HIF1α activity in WT and Irf3_P10A BMDCs and then challenged them with VSV. VSV-induced *Ifn-β* and *Ifit1* expression was still largely lower in Irf3_P10A BMDCs than in WT BMDCs (Supplementary Fig. 15j), indicating that HIF1α does not mediate the difference between WT and Irf3_P10A mice in response to viral infection.

To document whether viral infection could alter hypoxic conditions in mice, we examined HIF1α protein levels by immunofluorescence staining. Both VSV and HSV-1 infection increased HIF1α protein levels in the lungs of WT and Irf3_P10A mice upon viral infection for 24 h (Supplementary Fig. 16). These data suggest that a hypoxic state is achieved at the site of infection, which could lead to the attenuation of IRF3 activation by reducing EGLN1 activity, potentially facilitating viral immune escape.

Taken together, these data support a critical role for Irf3 proline-10 hydroxylation in potentiating antiviral innate immunity in vivo.

Based on these data, we propose a working model for the role of oxygen in antiviral innate immunity (Supplementary Fig. 17). Under normoxia and during viral infection, the activity of EGLN1 is sustained due to sufficiency of oxygen, leading to IRF3 hydroxylation at proline 10. Prolyl hydroxylation of IRF3 promotes IRF3 phosphorylation, dimerization and nuclear translocation and subsequent activation and, in turn, antiviral innate immune responses through the transcriptional transactivation of the type I interferon gene and subsequent interferon-stimulated genes. However, under hypoxia, the activity of EGLN1 is switched off, preventing prolyl hydroxylation of IRF3. This deficiency leads to a reduction in IRF3 phosphorylation, dimerization, nuclear translocation, and subsequent activation.

## Discussion

Molecular oxygen has a profound effect on organisms by providing energy during aerobic respiration and by altering the configuration and function of nucleic acids, sugars, lipids, proteins, and metabolites. Therefore, oxygen can influence inflammation and immunity through various mechanisms. However, due to the lack of feasible approaches to directly test the effect of oxygen on inflammation and immunity, how oxygen affects immunity is largely unclear. In contrast, the impact of hypoxia on inflammation and immunity has been recognized due to increased oxygen consumption by highly metabolically active inflamed resident cells and infiltrating immune cells[15]. Hypoxia affects inflammation and immunity by modulating immune cell proliferation, development, and effector functions, largely through transcriptional

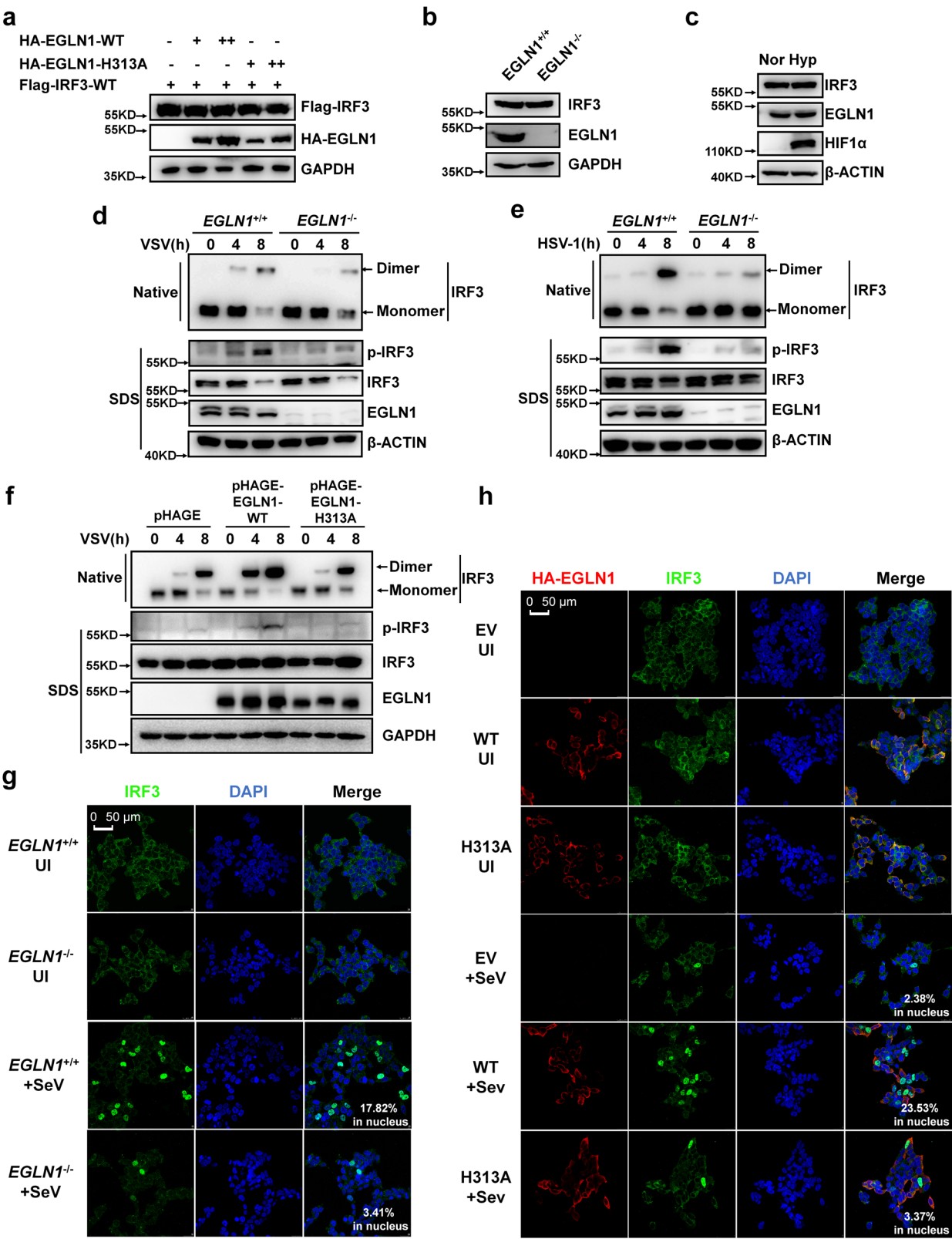

changes mediated by hypoxia-inducible factors (HIFs)[15]. However, whether and how oxygen can directly affect innate immune responses, in particular by influencing IRF3 signaling, remain unknown. In this study, we identified EGLN1 as an important mediator of the oxygen activity on antiviral innate immunity. Since EGLN1 is a key oxygen sensor and its enzymatic activity is dependent on oxygen, thus, our findings do not only directly link oxygen to antiviral innate immunity,

but also provide an ideal model to explain the mechanism by which oxygen acts on antiviral innate immunity.

Post-translational modifications (PTMs) of IRF3 cause either enhancement or suppression of IRF3 activation. IRF3 activation must undergo a sequential process, including phosphorylation, dimerization and subsequent nuclear translocation[29]. PTMs affect IRF3 activation at each of these steps. In this study, we found that IRF3 prolyl

**Fig. 7 | EGLN1 enhances IRF3 phosphorylation, dimerization, and nuclear translocation. a** Immunoblotting of the indicated protein expression in HEK293T cells transfected with the plasmid expressing Flag-tagged IRF3 together with increasing amounts of HA-*EGLN1* or HA-*EGLN1*-H313A. **b** Immunoblotting of the indicated protein expression in *EGLN1*[+/+] or *EGLN1*[−/−] H1299 cells. **c** Immunoblotting of the indicated protein expression in H1299 cells under normoxia (Nor) (21% $O_2$) and hypoxia (Hyp) (1% $O_2$) for 4 h. **d** *EGLN1*[+/+] and *EGLN1*[−/−] THP-1 cells were infected with VSV for the indicated times, and the cell lysates were analyzed by immunoblotting for monomeric (Monomer) and dimeric (Dimer) IRF3 (top; native-PAGE); phosphorylated IRF3 (p-IRF3), total IRF3, EGLN1, and β-ACTIN (bottom; SDS-PAGE). **e** *EGLN1*[+/+] and *EGLN1*[−/−] THP-1 cells were infected with HSV-1 for the indicated times, and the cell lysates were analyzed by immunoblotting for monomeric (Monomer) and dimeric (Dimer) IRF3 (top; native-PAGE); phosphorylated IRF3 (p-IRF3), total IRF3, EGLN1, and β-ACTIN (bottom; SDS-PAGE). **f** *EGLN1*[−/−] H1299 cells stably expressed pHAGE empty vector, wild-type EGLN1 (WT) or the enzymatically inactive mutant (H313A) by lentivirus were infected with VSV for the indicated times, and the cell lysates were analyzed by immunoblotting for monomeric (monomer) and dimeric (dimer) IRF3 (top, native-PAGE); phosphorylated IRF3 (p-IRF3), total IRF3, EGLN1, and GAPDH (bottom; SDS-PAGE). **g** *EGLN1*[+/+] or *EGLN1*[−/−] HEK293T cells were infected without (UI) or with SeV for 8 h and confocal microscopy image of endogenous IRF3 was detected by immunofluorescence staining using anti-IRF3 antibody. Scale bar = 50 μm. **h** *EGLN1*[−/−] HEK293T cells were transfected with the HA empty vector (EV), the plasmid expressing HA-tagged wild-type *EGLN1* (WT) or the plasmid expressing the enzymatically inactive mutant (H313A), followed by infected without (UI) or with SeV for 8 h. Confocal microscopy image of endogenous IRF3 was detected by immunofluorescence staining using anti-IRF3 antibody and HA-tagged EGLN1 or its mutant (H313A) was detected by immunofluorescence staining with anti-HA antibody. Scale bar = 50 μm. Data in (**a**–**h**) are representative from three independent experiments. See also Supplementary Fig. 12. Source data are provided as a Source data file.

hydroxylation promotes IRF3 phosphorylation, dimerization, and nuclear translocation, confirming the role of *EGLN1* in enhancing IRF3 function. Notably, the hydroxylation site of IRF3 by *EGLN1* is located at its N-terminus (proline 10), whereas its major phosphorylation sites of IRF3 are situated at the C-terminus[30,67–69]. How IRF3 P10 hydroxylation affects IRF3 phosphorylation to enhance its transcriptional activity remains a mystery. The IFN-β enhanceosome model suggests that the proline-10 region in the N-terminus of IRF3 is critical for the formation of a hydrogen bond in IRF3:DNA structure of the IFN-β enhanceosome[70]. Thus, the hydroxylation of IRF3 at proline 10 may also affect the DNA binding ability of IRF3.

Here, our finding that oxygen can enhance antiviral innate immunity suggests that oxygen uptake may be benefit in patients with viral infection and that the hypoxic conditions resulting from viral infection may lead to further enhancement of viral infection due to the reduction of PHD2 enzymatic activity.

## Methods

### Ethics statement
The laboratory animal facility was accredited by the Association for Assessment and Accreditation of Laboratory Animal Care International (AAALAC), and all the animal procedures used in this study were approved by the Institutional Animal Care and Use Committee (IACUC) of the Institute of Hydrobiology, Chinese Academy of Sciences.

### Reagents and antibodies
See Supplementary Table 1.

### Mice
Mice were housed (12-h light/dark cycle, 22°–26 °C) and given unrestricted access to standard diet and tap water under specific pathogen-free conditions at the Animal Research Center of Wuhan University. Six to eight-weeks old mice were used in the experiments. Same sex littermates were randomly assigned to the experimental groups.

### Generation of *Egln1* conditional knockout mice
C57BL/6J-*Egln1*[em1cyagen] mice were obtained from Cyagen Biosciences, which were generated by CRISPR/Cas9-mediated genome editing. Two loxP sites flank the exons 2 and 3 of the mouse *Egln1* gene. Cre-ER mice (B6.129-Gt(ROSA)26Sor[tm1(cre/ERT2)Tyj]) (hereafter referred to as Cre-ER) originally obtained from The Jackson Laboratory were kindly provided by Dr. Bo Zhong (Wuhan University). *Egln1*[fl/+] mice were crossed with Cre-ER mice to generate Cre-ER *Egln1*[fl/+] mice. After crossing, the Cre-ER *Egln1*[+/+] and Cre-ER *Egln1*[fl/fl] littermates were selected and used for further assays. The genotyping primers are listed in Supplementary Table 2.

To achieve conditional knockout of *Egln1*, 6-week-old Cre-ER *Egln1*[+/+] and Cre-ER *Egln1*[fl/fl] mice were injected intraperitoneally with tamoxifen (80 μg /per g body weight, dissolved in corn oil) (Sigma, #T5648) for five consecutive days. After 7 days without treatment, mice were either euthanized to test the knockout efficiency or infected with VSV. To knock out *Egln1* in cultured cells, Cre-ER *Egln1*[+/+] and Cre-ER *Egln1*[fl/fl] cells were treated with 4-hydroxytamoxifen (4OH-tamoxifen) (1 μM) (Sigma, H6278) for three days. Cells were reseeded into culture dishes or plated in 4-hydroxytamoxifen-free medium and allowed to rest for 24 h, followed by infection with VSV or HSV-1.

### Generation of *Irf3*-P10A mice
A C57BL/6 mouse model with a point mutation (P10A) in the mouse Irf3 locus was generated by CRISPR/Cas9-mediated genome editing and obtained from the Cyagen Biosciences. Briefly, exon 2 of the *Egln1* gene was selected as the target site and guide RNAs (5′-ACCGCGGATT TTGCCCTGGCTGG-3′) were obtained by in vitro transcription and purification. Cas9 mRNA, gRNA and donor oligo containing P10A (CCC to GCG) mutation sites (with targeting sequence, flanked by 140 bp homologous sequences combined on both sides) were co-injected into fertilized mouse eggs to generate targeted knockin offspring. F0 founder animals were identified by PCR followed by sequence analysis and bred to wild-type mice to test germline transmission and generation of F1 animals. Genotyping primers are listed in Supplementary Table 2. The Irf3-P10A mice were backcrossed to a C57BL/6 background for seven generations prior to this study.

### Zebrafish
Zebrafish were bred and maintained in a recirculating water system according to standard protocols. *The egln1a*[ihb1228/ihb1228] (*egln1a*[−/−]) (https://zfin.org/ZDB-ALT-180803-3), *egln1b*[ihb1229/ihb1229] (*egln1b*[−/−]) (https://zfin.org/ZDB-ALT-180803-4) and *egln1a*[ihb1228/ihb1228] *egln1b*[ihb1229/ihb1229] (*egln1a*[−/−]*egln1b*[−/−]) double mutant zebrafish were previously described (Fan et al.[62]). Tg(ifnφ1:mCherry) and Tg(lyz:DsRed2) zebrafish were obtained from the China Zebrafish Resource Center, National Aquatic Biological Resource Center (NABRC-CZRC).

### Cell culture
HEK293T, H1299, THP-1 and Vero cell lines were originally obtained from ATCC (American Type Culture Collection). The 786-O cell line was a gift from Dr. William Kaelin. RCC4 cell line was a gift from Dr. Peter Ratcliffe. *Irf3*[−/−]*Irf7*[−/−] MEF cells were provided by Dr. Bo Zhong (Wuhan University, China). Cells were grown at 37 °C in a humidified incubator containing 5% $CO_2$. HEK293T, H1299, RCC4 and Vero cells were cultured in Dulbecco's modified Eagle's medium (DMEM) (Viva-Cell) with 10% fetal bovine serum (FBS) (VivaCell). THP-1 and 786-O cells were cultured in RPMI 1640 medium (VivaCell) supplemented

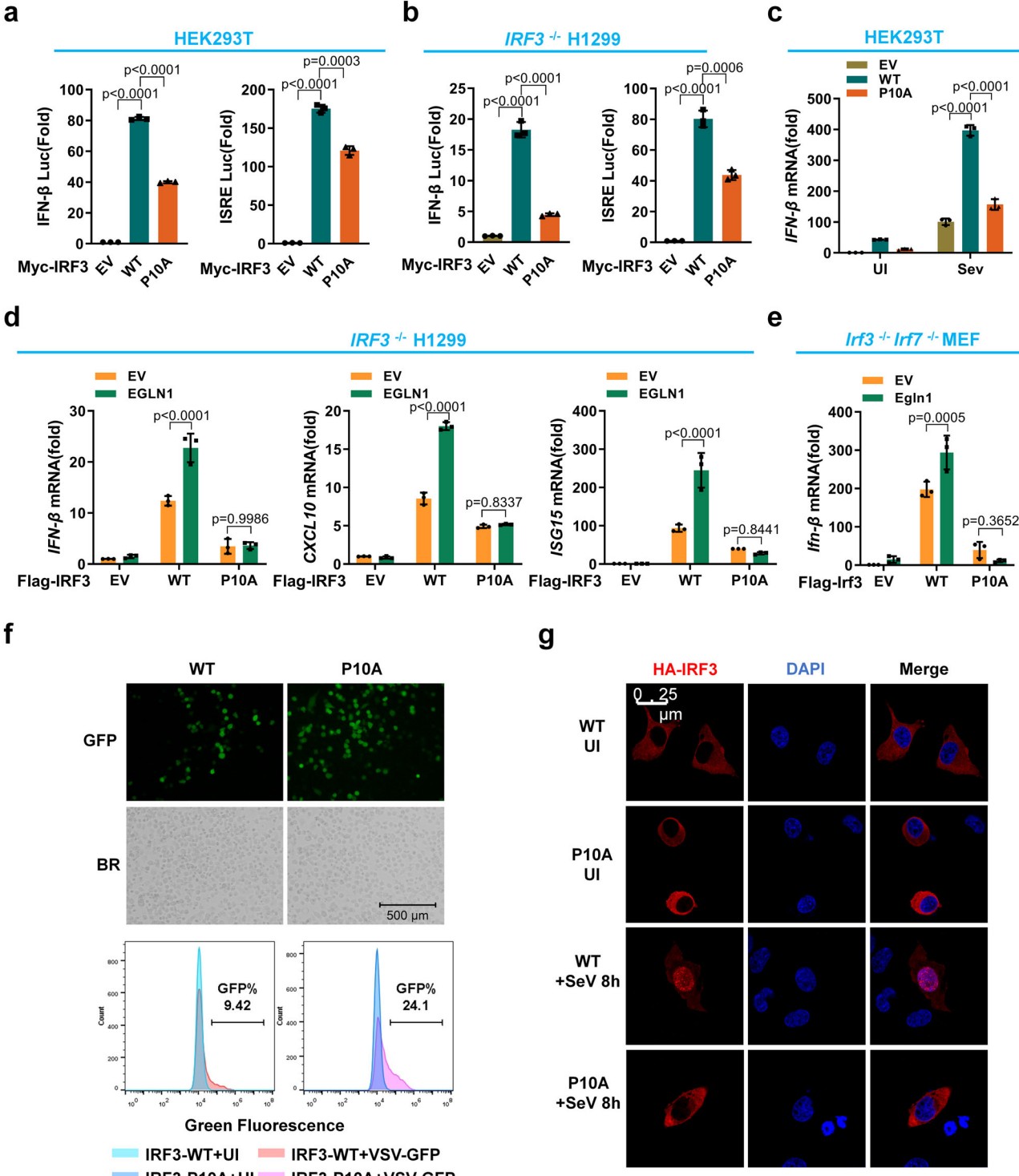

with 10% FBS. Mouse embryonic fibroblasts (MEFs) were maintained in DMEM supplemented with sodium pyruvate (110 mg/l), 10% FBS, 1× non-essential amino acids (VivaCell) and 1% penicillin–streptomycin (VivaCell).

Bone marrow cells were isolated from mouse femurs. Cells were grown at 37 °C in a humidified incubator containing 5% $CO_2$. The cells were cultured in DMEM containing 10% FBS, 1% streptomycin–penicillin, and 10 µM β-mercaptoethanol (Sigma) with M-CSF (10 ng/ml, Peprotech) for BMDM differentiation or GM-CSF (20 ng/ml, Peprotech) for BMDC differentiation, respectively. On day 6, the cells were reseeded and used for subsequent experiments.

Primary human peripheral blood mononuclear cells (PBMC) (#CP-H158) or leukocytes (#CP-H030) purchased from Procell (Wuhan, China) were isolated from whole blood of healthy donors by density gradient centrifugation. Cells were cultured in special culture media (#CM-H158) or (#CM-H030) containing 10% FBS, 1% streptomycin-penicillin, respectively. Cells were grown at 37 °C in a humidified incubator containing 5% $CO_2$.

Epithelioma papulosum cyprini (EPC) cells (originally obtained from the American Type Culture Collection) were cultured in Medium 199 Earle's Salts Base medium (VivaCell) containing 10% FBS, 1% streptomycin–penicillin at 28 °C in a humidified incubator containing 5% $CO_2$.

**Fig. 8 | Hydroxylation of IRF3 at proline 10 enhances IRF3 activation and nuclear translocation in cellular antiviral immune responses. a** IFN-β promoter activity and ISRE reporter activity in HEK293T transfected with the Myc empty vector (EV) or the plasmid expressing wild-type IRF3 (WT) or its mutant (P10A). **b** IFN-β promoter activity and ISRE reporter activity in *IRF3*<sup>-/-</sup> H1299 cells transfected with the Myc empty vector or the plasmid expressing wild-type IRF3 or its mutant (P10A). **c** qPCR analysis of *IFN-β* mRNA in HEK293T transfected with the empty vector or the plasmid expressing wild-type IRF3 (WT) or its mutant (P10A), followed by infected without (UI) or with SeV for 8 h. **d** qPCR analysis of *IFN-β*, *CXCL10* and *ISG15* mRNA in *IRF3*<sup>-/-</sup> H1299 transfected with the empty Flag vector or the plasmid expressing Flag-tagged wild-type IRF3 or its mutant (P10A), together with the empty Myc vector or the plasmid expressing Myc-tagged EGLN1 (EGLN1). **e** qPCR analysis of *Ifn-β* mRNA in *Irf3* and *Irf7*-deficient MEF (*Irf3*<sup>-/-</sup>*Irf7*<sup>-/-</sup>) transfected with the empty Flag vector or the plasmid expressing Flag-tagged wild-type Irf3 or

its mutant (P10A), together with the empty Myc vector or the plasmid expressing Myc-tagged Egln1 (Egln1). **f** *IRF3*<sup>-/-</sup> H1299 were transfected with the plasmid expressing Flag-tagged wild-type IRF3 or its mutant (P10A) for 24 h, followed by infection without or with VSV-GFP viruses for 12 h, and viral infectivity was detected by fluorescence microscopy or flow cytometry analysis (bottom panels). **g** *IRF3*<sup>-/-</sup> H1299 were transfected with the plasmid expressing HA-tagged wild-type IRF3 or P10A mutant (P10A), followed by infection without or with SeV for 8 h. Confocal microscopy image of exogenous IRF3 was detected by immunofluorescence staining with anti-HA antibody. Scale bar = 25 μm. BR, bright field. Data in (**a**, **b**) are presented as mean ± S.D., two-tailed Student's *t* test; *n* = 3 biological independent experiments. Data in (**c**–**e**) are presented as mean ± S.D., two-way ANOVA; *n* = 3 biological independent experiments. Data in (**f**, **g**) are representative from three independent experiments. See also Supplementary Fig. 13. Source data are provided as a Source data file.

## Viruses

Sendai virus (SeV) (provided by Dr. Bo Zhong; Wuhan University, China) was used at a final concentration of 100 hemagglutinating units/ml. VSV and HSV-1-GFP viruses (provided by Dr. Hong-Bing Shu; Wuhan University, China), VSV-GFP viruses (provided by Dr. Mingzhou Chen; Wuhan University, China), and HSV-1 (provided by Dr. Chunfu Zheng; Fujian Medical University, Fuzhou, China) were propagated and tested by plaque assay on Vero cells. GCRV (genotype II) were gifts from Lingbing Zeng (Yangtze River Fisheries Research Institute, Chinese Academy of Fishery Sciences, Wuhan, China). Spring viremia of carp virus (SVCV) (provided by Dr. Shun Li; Institute of Hydrobiology, Chinese Academy of Sciences, China) was grown in EPC cells and viral titers were determined by a 50% tissue culture infectious dose (TCID$_{50}$) assay on EPC cells.

## Virus plaque assay

Medium containing VSV or HSV-1 virus was diluted and used to infect confluent monolayer Vero cells cultured in 12-well plates. One hour after infection, the cells were washed three times with pre-warmed PBS followed by incubation with DMEM containing 2% methylcellulose for 48 h. The cells were fixed with 4% paraformaldehyde for 15 min and stained with 1% crystal violet for 30 min. Finally, the plaques were counted, averaged, and multiplied by the dilution factor to determine the viral titer as PFU/ml.

## Luciferase reporter assay

Cells grown in 24-well plates were transfected with different amounts of plasmids using VigoFect (Vigorous Biotech, Beijing, China), and pCMV-*Renilla* as an internal control. After transfection for 18–24 h, the luciferase activity was determined using the dual-luciferase reporter assay system (Promega, #E1960). Data were normalized to *Renilla* luciferase. Data are experssed as mean ± standard deviation (S.D.) of a representative experiment performed in triplicate based on three independent experiments.

## Quantitative real-time RT-PCR (qPCR) assay

Total RNA was extracted using RNAiso Plus (TaKaRa Bio., Beijing, China) according to the manufacturer's protocol. cDNA was synthesized using the Revert Aid First Strand cDNA Synthesis Kit (Thermo Scientific, Waltham, MA, USA). MonAmp<sup>TM</sup> SYBR® Green qPCR Mix (high ROX) (Monad Biotech, Shanghai, China) was used for qRT-PCR assays. Primers for qRT-PCR assays are listed in Supplementary Table 2.

## CRISPR-Cas9 knockout cell lines

To generate *EGLN1*-, *IRF3*-, *HIF1β*- and *Egln1*-knockout cell lines, sgRNA sequences corresponding to the indicated target sites were ligated into LentiCRISPRv2 plasmid and then co-transfected with viral packaging plasmids (psPAX2 and pMD2G) into HEK293T cells. The medium was changed at 8 h post-transfection, and the viral supernatant was collected 48 h later. Target cells were infected with viral supernatant and selected with 1 μg/ml puromycin for 2 weeks.

## Lentivirus-mediated gene transfer

Cells were transfected with pHAGE-EGLN1, pHAGE-EGLN1-H313A, or the empty vector together with the packaging vectors pSPAX2 and pMD2G. The medium was changed to fresh complete medium (10% FBS, 1% streptomycin–penicillin) after 8 h. Forty hours later, the supernatants were harvested. Target cells were infected with viral supernatant and selected with 1 μg/ml puromycin for 2 weeks.

## Flow cytometry analysis for VSV-GFP or HSV-1-GFP infected cells

The cells were infected with VSV-GFP or HSV-1-GFP viruses for the indicated time, and then harvested and washed with PBS. Cells were then counted at 10,000 cells and analyzed using Beckman CytoFLEXS. Data were analyzed and generated using FlowJo software.

## Cell apoptosis analysis

FITC Annexin V Apoptosis Detection Kit I (BD Pharmingen, #556547) was used to measure the effect of FG4592 (0–100 μM) on cell apoptosis of H1299 or THP-1 cells according to the manufacturer's instructions. Apoptotic cells were detected using Beckman CytoFLEXS and the data were analyzed using CytExpert software.

## Immunoblotting and immunoprecipitation

Total cell protein was extracted with RIPA buffer containing 50 mM Tris (pH 7.4), 1% Nonidet P-40, 0.25% sodium deoxycholate, 1 mM EDTA (pH 8.0), 150 mM NaCl, 1 mM NaF, 1 mM PMSF, 1 mM Na3VO4, a 1:100 dilution of phosphatase inhibitor cocktail (Cell Signaling Technology, #5870 S), and a 1:100 dilution of protease inhibitor mixture (Bimake, #B14001). Cell lysates were separated by SDS-PAGE, transferred to a polyvinylidene difluoride (PVDF) membrane (Millipore), blocked with 5% (w/v) nonfat milk, probed with the indicated primary antibodies and corresponding secondary antibodies, visualized using ECL Western blotting detection reagent (Millipore) and photographed using a Fuji Film LAS4000 mini-luminescent image analyzer. Anti-Flag antibody-conjugated agarose beads (Sigma-Aldrich, #A2220), Anti-Myc antibody-conjugated agarose beads (Sigma-Aldrich, #A7470), and anti-HA antibody-conjugated agarose beads (Sigma-Aldrich, #A2095) were used for the exogenous co-immunoprecipitation assay. Protein G Sepharose (GE HealthCare Company, #17-0618-01) was used for the endogenous co-immunoprecipitation assay. Image J software (National Institutes of Health) was used to quantify protein levels based on the band density obtained by Western blot analysis.

## Immunofluorescent confocal microscopy

Cells grown on glass coverslips were fixed with 4% paraformaldehyde in PBS for 20 minutes, permeabilized with 0.1% Triton X-100 and blocked with 1% bovine serum albumin. Cells were then stained with the indicated primary antibodies (anti-HA, 1:1000, Cell Signaling Technology, #3724; anti-Flag, 1:1000, Sigma-Aldrich, #F1804; anti-IRF3, 1:500, Cell Signaling Technology, #11904) followed by incubation with fluorescence dye-conjugated secondary antibodies. Cell nuclei

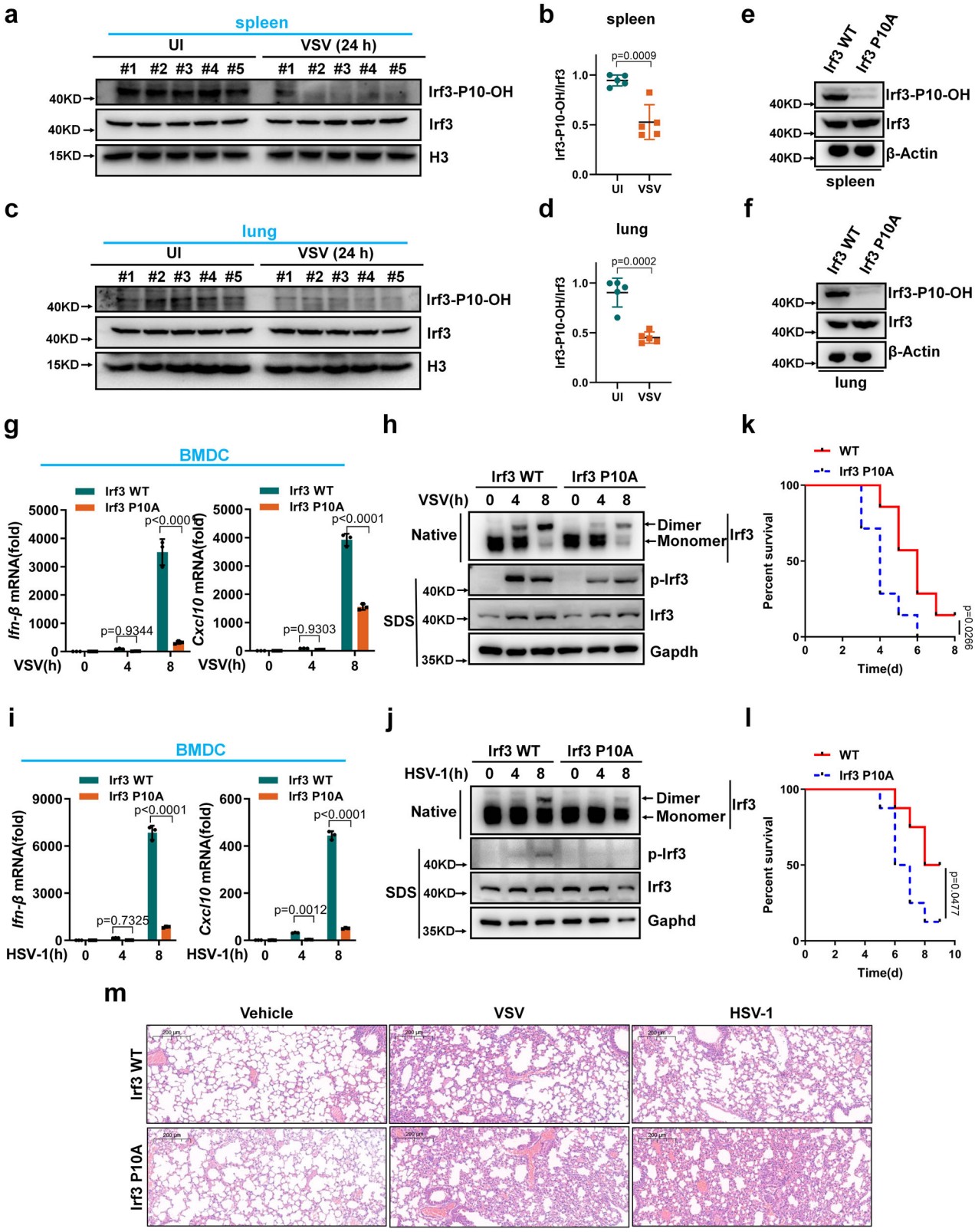

were counterstained with DAPI (Thermo Fisher). Cells were imaged using a Leica laser scanning confocal microscope.

**GST pull-down assay**

GST-tagged IRF3 and His-tagged EGLN1 were expressed in *Escherichia coli* (BL21), respectively. GST resin (Novagen) and His resin (Novagen) were used for protein purification. GST (1 μg) or GST-IRF3 (1 μg) was incubated with His-EGLN1 (1 μg) overnight (4 °C). The association of

GST-IRF3 with His-EGLN1 was detected by immunoblotting with anti-EGLN1 and anti-IRF3 antibodies. GST and GST-IRF3 proteins were stained with Coomassie blue.

**Proximity ligation assay**

H1299 cells were fixed and permeabilized as described above. Proximity ligation assay (PLA) was performed according to the manufacturer's instructions for the Duolink In Situ Red Starter Kit Mouse/

**Fig. 9 | Irf3 prolyl hydroxylation deficiency attenuates antiviral innate immunity in mice. a** Irf3 prolyl hydroxylation in spleen of wild-type mice infected without (UI) or with VSV for 24 h. **b** Quantitation of Irf3 prolyl hydroxylation in (**a**). **c** Irf3 prolyl hydroxylation in lung of WT mice infected without or with VSV for 24 h. **d** Quantitation of Irf3 prolyl hydroxylation in (**c**). **e, f** Proteins in spleen (**e**) and lung (**f**) of *Irf3*_P10A mutant and WT littermates (*Irf3*_P10A and *Irf3*-WT). **g** qPCR analysis of *Ifn-β* and *Cxcl10* mRNA in WT or *Irf3*_P10A mutant BMDCs (*Irf3*-WT or *Irf3*_P10A) infected with VSV. **h** WT or *Irf3*_P10A mutant BMDCs (*Irf3*-WT or *Irf3*_P10A) were infected with VSV for the indicated times, and the cell lysates were analyzed by immunoblotting. **i** qPCR analysis of *Ifn-β* and *Cxcl10* mRNA in WT or *Irf3*_P10A-mutant BMDCs (*Irf3*-WT or *Irf3*_P10A) infected with HSV-1. **j** WT or *Irf3*_P10A mutant BMDCs (*Irf3*-WT or *Irf3*_P10A) were infected with HSV-1 for the indicated times, and the cell lysates were analyzed by immunoblotting. **k** Survival (Kaplan-Meier curve)

of WT mice and *Irf3*_P10A mutant mice injected intraperitoneally with a high dose of VSV ($5 \times 10^7$ PFU per mouse). Statistical analysis was performed using the log-rank test (*n* = 7 for each group). **l** Survival of WT mice and *Irf3*_P10A mutant mice injected intraperitoneally with a high dose of HSV-1 ($5 \times 10^7$ PFU per mouse). Statistical analysis was performed using the log-rank test (*n* = 8 for each group). **m** H & E stained images of lung sections from WT or *Irf3*_P10A mutant mice (*Irf3*-WT or *Irf3*_P10A) injected intraperitoneally with PBS (phosphate buffer saline), VSV ($5 \times 10^7$ PFU per mouse) or HSV-1 ($5 \times 10^7$ PFU per mouse) for 24 h. Data in (**a, c, e, f, h, j, m**) are representative from three independent experiments. Data in (**b, d**) are presented as mean ± S.D., two-tailed Student's *t* test; *n* = 5 biological independent samples. Data in (**g, i**) are presented as mean ± S.D., two-way ANOVA; *n* = 3 biological independent experiments. See also Supplementary Figs. 14–16. Source data are provided as a Source data file.

Rabbit (Duolink, Sigma-Aldrich, DUO92101). Cells were incubated with primary antibodies (anti-HA, 1:1000, Cell Signaling Technology, #3724; anti-IRF3, 1:200, Cell Signaling Technology, #F4302) for 1 h at room temperature and nuclei were counterstained with DAPI (Thermo Fisher). Red fluorescent spots were detected using a Leica laser scanning confocal microscope.

### Enzyme-linked immunosorbent assay (ELISA)
Concentrations of IFN-β in THP-1 cell supernatants were measured using ELISA Kits (R&D Systems, #DIFNB0) according to the manufacturer's protocol. Ifn-β concentrations in BMDC supernatants and sera were measured using ELISA Kits (Bio legend, #439407) according to the manufacturer's instructions.

### Identification of IRF3 hydroxylation site(s) by mass spectrometry
HEK293T cells were transfected with Flag-IRF3 plasmids. Cell lysate was immunoprecipitated with anti-Flag Ab–conjugated agarose beads overnight. Immunoprecipitated IRF3 proteins were subjected to 8% SDS-PAGE gel, and IRF3 bands were excised from the gel and analyzed by mass spectrometry in ProteinGene Biotech (Wuhan, Hubei, China). The MaxQuant software was used for all MS/MS spectra analyses against the human protein database (https://www.ncbi.nlm.nih.gov/protein) combined with the reverse decoy database and common contaminants. Carbamidomethylation (Cys) was used as the fixed modification. Dynamic modifications included deamidation (Asn/Gln), oxidation (Met), acetylation (protein N-terminal) and hydroxylation (Pro).

The digested peptides were dissolved in 0.1% formic acid, separated on an online nanoflow EASY-nLC 1200 system with a 75 μm × 15 cm analytical column (C18, 3 μm, Thermo Fisher Scientific) and then analyzed on a Q Exactive HF-X mass spectrometer (Thermo Fisher Scientific). Peptides were eluted with a gradient of solvent B (0.1% formic acid in 80% acetonitrile) increasing from 4% to 6% in 1 min, 6% to 10% in 2 min, 10% to 25% in 40 min, 25% to 45% in 10 min, climbing to 100% in 0.5 min, and then held at 100% for 6.5 min, all at a constant flow rate of 300 nl/min. The mass spectrometer was operated in data dependent acquisition mode with full scans (*m/z* range of 350–1800) at 60,000 mass resolution using an automatic gain control setpoint of 3e6. The top 20 most intense precursor ions were selected for subsequent MS/MS fragmentation by higher energy collision dissociation with normalized collision energy of 28% and analyzed in the Orbitrap at 15,000 resolution. The dynamic exclusion was set to 25 s and the precursor ion isolation width to 1.6 *m/z*. The maximum injection times were 20 ms and 50 ms for both MS and MS/MS, respectively. The intensity threshold was set to 5000.

### Generating site specific hydroxylation antibody
An IRF3 P10 site-specific hydroxylation antibody (anti-IRF3-P10-OH) was generated by using a human IRF3 hydroxylated peptide [C-KPRIL(P-OH)WLVSQLD] as an antigen. After purification of antibodies

with excess unmodified peptide (C-KPRILPWLVSQLD), antibodies recognizing site-specific hydroxylation were enriched by biotin-labeled IRF3-hydroxylated peptides. The specificity of the anti-IRF3-P10-OH antibody was verified by dot blot.

### Expression and purification of GST-tagged proteins
Human *EGLN1*, human *IRF3*, and the ODD domain of human HIF1α (400-575 aa) were subcloned into the pGEX-4T vector. Recombinant GST-EGLN1, GST-IRF3, GST-HIF1α-ODD (400-575 aa) and GST protein were produced by transforming *Escherichia coli* BL21 (DE3) with pGEX-EGLN1, pGEX-IRF3, pGEX-HIF1α (400-575 aa) or pGEX-4T (GST protein control). Cultures were grown at 37 °C to 0.8 OD, protein expression was induced for 12–16 h with 0.1 mM IPTG at 16 °C with vigorous shaking. Recombinant proteins were purified from harvested pellets using the BeyoGold™ GST-tag purification kit (Beyotime, #P2262). Recovery and yield of the desired proteins were confirmed by analyzing 10 μl of beads by Coomassie blue staining, and quantified against BSA standards. To avoid interference of the GST tag in the subsequent in vitro hydroxylation reaction, thrombin (Beyotime, #ST1699) (0.2 U) was added and incubated at room temperature for 2 h to cleave the GST tag from the GST-HIF-ODD (400-575aa) and GST-IRF3 fusion proteins.

### In vitro hydroxylation assay
Assays were performed according to the previously described protocol with some modifications[65]. Briefly, purified HIF1α-ODD (400–575aa) or IRF3 (6 μg) was mixed in a 50 μl reaction volume with 50 mM HEPES (pH 7.4), 2000 U/μl catalase, 200 μM FeSO₄, 5 mM ascorbic acid, 200 μM α-ketoglutarate (α-KG), and 1 μg of purified recombinant GST-EGLN1 protein or 1 μg of GST protein as a control. To validate IRF3 hydroxylation by GST-EGLN1 by immunoblotting, we designed two additional controls, in which the reactions did not contain either FeSO₄ or α-ketoglutarate. After incubation for 2 h at 37 °C, the reactions were subjected to mass spectrometry analysis or immunoblotting with anti-HIF-OH antibody or anti-IRF3-P10-OH antibody.

For mass spectrometry analysis, the pFind software (version 3.1) (http://pfind.org/software/pFind/index.html) was used for all MS/MS spectra analyses against the human protein database (https://www.ncbi.nlm.nih.gov/protein) combined with the reverse decoy database and common contaminants. Two missed cleavages were allowed for trypsin and the open search algorithm in pFind was used. Hydroxylation (P) was also set as a variable modification. Precursor and fragment ion mass tolerances were 20 ppm and 20 ppm, respectively. The minimum peptide length was set to 6 while the estimated false discovery rate threshold for peptides and proteins was set to a maximum of 1%.

### FG4592 treatment and viral infection
For FG4592 (Selleck, #S1007) treatment of cells, H1299 or THP-1 cells were treated with FG4592 (0–100 μM) for 24 h, followed by cell apoptosis analysis; or the cells were treated with FG4592 (0–100 μM)

for 6 h, followed by virus infection and then qPCR or flow cytometry analysis.

Mice treated with FG4592 were approximately 6–8 weeks old mice (male). FG4592 was prepared as a 10 mg/ml solution in vehicle (5% DMSO, 40% PEG 300, 5% Tween 80 and 50% ddH$_2$O). Mice were treated with 10 mg/kg FG-4592 or vehicle alone by oral gavage twice daily. Mice were then infected with VSV ($5 \times 10^7$ PFU per mouse) by intraperitoneal injection followed by the indicated analysis.

For FG4592 treatment of zebrafish, wild-type zebrafish larvae (3 days post-fertilization, dpf) or Tg(ifnφ1:mCherry) zebrafish larvae (8 dpf) in egg water were treated with FG4592 (10 μM) for 24 h, followed by SVCV infection and indicated analysis.

## PX478 treatment

For PX478 treatment of cells, PX478 powder (Selleck, #S7612) was dissolved in DMSO (100 mM, stock solution). H1299 cells or BMDCs were treated with PX478 (10 μM) followed by virus infection and qPCR analysis.

## RNA sequencing

Total RNA from cells infected with or without VSV was purified using the RNeasy Mini Kit (QIAGEN NO. 74104). Sequencing libraries were generated using the NEB Next Ultra RNA Library Prep Kit for Illumina (NEB, USA, #E7530L) accordingly to the manufacturer's protocol, and index codes were added to assign sequences to each sample.

Briefly, mRNA was purified from total RNA using poly-T oligo-attached magnetic beads. Fragmentation was performed using divalent cations at elevated temperature in NEB Next First Strand Synthesis Reaction Buffer(5X). First strand cDNA was synthesized using random hexamer primers and M-MuLV reverse transcriptase (RNase H). Second strand cDNA was then synthesized synthesis using DNA Polymerase I and RNase H. Remaining overhangs were converted to blunt ends via exonuclease/polymerase activities. After adenylating the 3′ ends of the DNA fragments, NEB Next Adapters with a hairpin loop structure were ligated to prepare for hybridization. To preferentially select cDNA fragments of 370–420 bp in length, the library fragments were purified using the AMPure XP system (Beverly, USA). Then 3 μl of USER Enzyme (NEB, USA) was used with the size selected, adapter ligated cDNA at 37 °C for 15 min followed by 5 min at 95 °C before PCR. PCR was then performed using Phusion High-Fidelity DNA Polymerase, Universal PCR primers and Index (X) primers. Finally, the PCR products were purified (AMPure XP system) and library quality was assessed on the Agilent 5400 system (Agilent, USA) and quantified by qPCR (1.5 nM). The qualified libraries were pooled and sequenced on Illumina platforms using the PE150 strategy at Wuhan Benagen Technology Co., Ltd. (Wuhan, China).

Fastp (version 0.19.7) was used to perform basic statistics on the quality of the raw reads. The steps of data processing steps were as follows: Discard a paired read if one of the reads contains adapter contamination; Discard a paired read if more than 10% of bases are uncertain in one of the reads; Discard a paired read if the proportions of uncertain bases in one of the reads are greater than 10%.

The clean reads were then mapped to the mouse genome using HISAT. The RSEM package was used to calculate gene expression levels for each sample, expressed as fragments per kilobase of transcript per million fragments mapped (FPKM). Gene expression levels in stressed samples were compared with those in control samples to identify the differentially expressed genes (DEGs). DEGs were detected based on the parameters described previously: Fold change ≥ 2.00 and Probability ≥ 0.8 with a significant false discovery rate-adjusted P value (FDR) < 0.05 based on the three biological replicates. The heat map of expression changes of the indicated genes between different samples was generated using the Multi Experiment Viewer (MeV) software. Gene Ontology (GO) and Kyoto Encyclopedia of Genes and Genomes (KEGG) enrichment analyses for the DEGs were performed using Cluster Profiler version 3.8.

## Viral infection in mice

For in vivo viral infection, age- and sex-matched (male) mice were infected with VSV ($1 \times 10^7$ PFU per mouse) or HSV-1 ($1 \times 10^7$ PFU per mouse) by intraperitoneal injection. Lungs from the control or virus-infected mice were harvested 24 h post-infection for histological analysis. The mRNA levels of the indicated targets in the spleen and lung were determined by qPCR assays. The hydroxylation of Irf3 in the spleen and lung was determined by Western blot using anti-IRF3-P10-OH antibody.

For the survival rate, mice were injected intraperitoneally with VSV ($1 \times 10^7$ PFU per mouse or $5 \times 10^7$ PFU per mouse) or HSV-1 ($5 \times 10^7$ PFU per mouse) and monitored for the indicated days.

## Lung histology

Lungs from the control or virus-infected mice were dissected, fixed in 10% phosphate-buffered formalin, embedded into paraffin, sectioned, stained with hematoxylin and eosin solution, and examined by light microscopy. Anti-HIF1α antibody (Novus, #NB100-134) was used for immunofluorescence staining and images were captured by fluorescence microscopy as described above.

## Viral infection in zebrafish

For SVCV infection of zebrafish larvae, 30 larvae (3 dpf) were pooled in a disposable 60-mm cell culture dish filled with 4 ml of egg water and 1 ml of SVCV solution (~$6 \times 10^7$ TCID$_{50}$ per milliliter). The infection procedure was performed at 28 °C and monitored every 6 h over a 24-h period to assess mortality.

For GCRV II infection of zebrafish larvae, 30 larvae (3 dpf) were pooled in a disposable 60-mm cell culture dish filled with 4.7 ml of egg water and 0.3 ml of GCRV II. The infection procedure was performed at 28 °C and monitored every 2 h over a 26-h period to assess mortality.

For virus injection into adult zebrafish, 5-month-old zebrafish were injected intraperitoneally with 10 μl SVCV (~$6 \times 10^7$ TCID$_{50}$ per milliliter) or cell culture medium as a control. The mRNA levels of the indicated targets in the liver and kidney were determined by qPCR assays.

## Statistics and reproducibility

No statistical methods were used to estimate sample size. Differences between experimental and control groups were determined by unpaired two-tailed Student's $t$ test (when two groups of data were compared) or two-way ANOVA analysis (when more than two groups of data were compared). Statistical analysis of survival curves was performed using the log-rank test (Mantel–Cox). Statistical analyzed data are expressed as mean ± standard deviation (S.D.). A $p$ value < 0.05 was considered statistically significant. Statistical analyses were performed using GraphPad Prism 8.0. Data were obtained from three independent reproducible experiments.

## Reporting summary

Further information on research design is available in the Nature Portfolio Reporting Summary linked to this article.

# Data availability

Raw mass spectrometry data have been deposited to the ProteomeXchange Consortium via iProX partner repository with the dataset identifiers PXD050566 and PXD048667. The original RNA-seq data were uploaded to the GEO Datasets (GEO accession GSE253181). Any additional information required to reanalyze the data reported in this paper is available from the corresponding author upon request. Source data are provided with this paper.

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

## Acknowledgements

We thank Cyagen Biosciences Inc. (Guangzhou, China) for assistance with generation of *Egln1* conditional KO mice and Irf3 P10A mutation mice. We thank Dr. Hong-Bing Shu, Bo Zhong, Mingzhou Chen, and Chunfu Zheng, Lingbing Zeng, Shun Li for providing reagents. We thank Yan Wang and Fang Zhou at the Core Facility of Institute of Hydrobiology for FACS and confocal microscope. We thank Dr. Mingkun Yang for mass spectrometry analysis. This work was supported by grants from the Strategic Priority Research Program of the Chinese Academy of Sciences (XDB0730300 to X.L. and XDA24010308 to W.X.), the National Key Research and Development Program of China (2022YFF1000302 to X. L. and 2018YFD0900602 to W.X.), NSFC (32273171 to X.L.), "Agricultural Biological Breeding-2030" major project (2023ZD04065 to X.L.), and the Natural Science Foundation of Hubei Province of China (2022CFA110 to X.L.).

## Author contributions

W.X. and X.L. conceived and designed the study, analyzed data, and wrote the manuscript. J.T. and Z.W. contributed to the most aspects of experiments; C.Z., H.D., X.S., G.Y., F.R., X.C., Q.L., S.J., W.L., H.Z., S.F., X.C. contributed to various aspects of experiments. J.G. provided guidance and analyzed data.

## Competing interests

The authors declare no competing interests.
