## [Peer Review File · Nature Communications]

Oxygen enhances antiviral innate immunity through maintenance of EGLN1-catalysed proline hydroxylation of IRF3REVIEWER COMMENTS

Reviewer #1 (Remarks to the Author):

Upon sensing of intracellular viruses either via their DNA or RNA, signaling cascades are initiated that eventually converge in the activation of the transcription factor IRF3, which then drives expression of type I Interferons and other pro-inflammatory cytokines. The activity of the interferon induction cascade is tightly regulated and post-translational modifications (PTM) play a crucial role.

In this manuscript Xing Liu and coworkers show that hypoxic conditions suppress induction of antiviral genes independent of HIF signaling. They show that under normoxia the prolyl hydroxylase EGLN1 catalyzes the hydroxylation of IRF3 at proline 10. This PTM enhances IRF3's activation, leading to an enhanced gene activation under normoxia, which is suppressed under hypoxia. Molecular studies are elegantly combined with in vivo studies in mice and zebrafish, using EglN1 deletion mutants that showed higher susceptibility to viral infection, confirming the in vitro phenotype and mechanism.

Overall, the data and conclusions of the manuscript are strong, relevant, highly interesting and convincing. The experiments are well conducted and controlled. Thus, the hypothesis is fully supported by the created data. I only have a few suggestions that may improve the current manuscript.

Major:

- The figures are a bit overbearing. I suggest to streamline the main figure panels focusing on the key information in the figures, and move e.g. repeats in different cell lines or alternative readouts to the supplements. This allows the reader to assess the data better and additionally facilitates the readability of figure information, which is currently occasionally very small. For example, the error bars, and individual points are barely discernable, e.g. Fig. 1b-d, f, g, i-k, m – q; Fig. 2b-d, f-l, k, m, o, q, r; Fig. 6b, d, k, g, h, i, j, q-x, axes in the flow cytometric analyses are very small (Fig. 2g, s), legends could be reassigned and images with increased resolution should be used in the following figures (e.g. in Fig. 3a the loxP site indicated green is hardly visible, b-m, o-q; Fig. 4f, s, u) and Fig. 3 and 5 containing Elgn1^{-/-}, ELGN1^{-/-}, and ELGN1^{+/+}, Elgn1^{+/+} and Elgn1^{fl/fl}.

- The authors nicely showed the detrimental effect of Elgn1^{-/-} in mice. However, the viral load (e.g. via plaque forming assay or qPCR) during the infection/at the timepoint of sacrificing would provide valuable insight. I also suggest to quantify the immunofluorescence/flow cytometry viral loads in Fig. 2g, s, 3c. In addition, I would suggest to add a statement whether the baseline of inflammation i.e. in the non-infected animals were similar in KO and WT mice. Do KO mice already start with lower baseline restriction factor/ISG expression (e.g. as a supplementary figure).

- The discussion part could be more comprehensive. The authors could discuss their findings in a physiological or clinically relevant context, this would highlight the importance of the displayed results. Further, the relevance of the identified system e.g. during infection which causes hypoxia could be discussed and approaching the question why it is in place. Related to this, are there any other known oxidational processes in other immune pathways?

Minor:

- Check for abbreviation completeness (e.g. Fig. 3 (Cre-ER, ER = ?) and remove

abbreviations where they are not necessary (e.g. Line 39-43).

- Please maintain similar abbreviations throughout the manuscript e.g. Irf3 vs IRF3.

- Fig. 3D: Move GFP band in the center of the depiction.

- The methods descriptions are brief and detail could be added. I further suggest to deposit large datasets on public databases instead of making them available only upon request (as currently stated in the manuscript).

Reviewer #2 (Remarks to the Author):

Liu Nature Communications

This is an interesting paper linking oxygen availability to innate immunity via the hydroxylation of IRF3. Since the field is riddled with non-reproducible hydroxylation targets, it is imperative that the authors tighten up the evidence that IRF3 is hydroxylated in cells and by EglN1 in vitro (specific comments below). They would also be well served to delete or repeat (in some cases with advice below) some of the experiments where the differences observed were very small and hence of uncertain biological relevance.

Major Comments:

1. On a proportional basis the induction of IFN-beta by EglN1 compared with EV in the VSV treated cells in Extended Data Fig. 7C (IRF3^{-/-} cells) seems as great (or greater) than, for example, in the IRF3^{+/+} cells in Extended Data Fig. 1B. I understand the former and not the latter achieved statistical significance but that could simply reflect the noisiness of 7C (i.e. a few more replicates and it might become significant).
2. The description of how Fig. 4F was done in Results and Legend is COMPLETELY inadequate. I can't even tell if it was an in vivo or in vitro assay. Overexpressed IRF3 or endogenous? I can't read the spectra (even after magnifying) and should include analysis of the corresponding synthetic peptides with P10 hydroxylated or not.
3. Fig. 4H can't be interpreted- it only says the antibody requires the proline. A better experiment would be plus or minus DMOG or plus or minus EglN.
4. Fig. 4L is unacceptable/not believable (difference too subtle). Better to reintroduce wild-type or mutant EglN1 into EglN1^{-/-} cells to get better signal to noise.
5. Fig. 4P-Q also very poor. I applaud the attempt to quantify but doesn't look robust or biologically relevant.
6. The differences in Fig. 5I and 5K are completely underwhelming.
7. I can't read the spectra (even after magnifying) in Fig. 4U and should include analysis of the corresponding synthetic peptides with P10 hydroxylated or not. For Figs. 4F and 4U might also be helpful to comment on percent hydroxylation.
8. The differences in Fig 6L and 6M are completely underwhelming.
9. A caveat is that the proline to alanine mutant is a non-conservative mutation that could affect function independently of hydroxylation. The use of FG4592 and EglN1^{-/-} cells is very helpful, however,

Minor Comments:

1. Line 77. Make no sense to write "However, how the hydroxylation of IRF3 function...." here because the authors haven't yet described the fact that IRF3 is hydroxylated.

2. I think the Fig. 1 legend should give a sense of how much VSV and how much poly I:C was added.
3. Line 96. I think the authors mean “consistent with these findings”, not “consistently”
4. Line 107. I think the authors might mean “rarely”, not “barely” and might want to be saying “can be” less than 1%.
5. Line 114. Should just describe the finding. “cope with the decrease” infers the biological rationale, which doesn’t really belong here.
6. How do the levels of exogenous EglN1 in Fig. 2A compare with endogenous levels??
7. The induction of IFNB and CXCL10 in extended data Fig. 1 is very modest (maybe because EglN1 is not limiting in these cells) and of uncertain biological relevance.
8. Line 135, etc. Once again, “consistently” is not the right word here (consistently could mean reproducibly here). Think the authors mean, consistent with these findings”
9. The Figure 2 legend should make clear whether the knockout cells are clonal or polyclonal.
10. Why introduce IFIT1 measurements in Fig 2O? And the effect of wild-type EglN1, although statistically significant, is unlikely to be biologically relevant.
11. Should acknowledge that this is not the first report of conditional EglN1 mice.
12. HIF stabilization usually lowers protein translation. How do you know the effect in Extended Data Fig. S5H is specific?
13. A caveat with Fig. 4A-C, of course, is the proteins are OVERproduced. Are the results at least specific to EglN1 relative to EglN2 and 3?
14. The description of Fig. 4E in the legend is inadequate (e.g. I guess the cells were also transfected to produce FLAG-IRF3 but that is not stated).
15. Extended Data S6A lacks specificity controls or information about protein concentrations.
16. The enhancement of IRF3 activity by EglN1 is very modest in Extended Data Fig. 8i. Wonder if would be more impressive in EglN1^{-/-} cells.
17. Fig. 6W and 6X are simply “negative data” without real controls to show HIF is downregulated and with a very questionable tool compound. I would probably delete.

Reviewer #3 (Remarks to the Author):

In this manuscript, Liu et al. discover that hypoxia condition suppresses innate immune signaling pathways. Mechanistically, they report that the suppression is EGLN1-dependent and HIF-1-independent. EGLN1 catalyzes proline 10 hydroxylation of IRF3 which enhances IRF3 phosphorylation and dimerization, in the presence of oxygen. The authors further provide in vivo data with recombinant mice/zebrafish and EGLN inhibitor to consolidate the positive regulation on innate immunity by EGLN1. Overall, the study is thoughtfully designed with convincing data. There are several aspects that need further explanation/exploration to substantiate the conclusions:

Major:

1. In figure 1, the only data supporting the HIF signaling-independence are Figure 1e, 1f, and 1g, while the other data merely demonstrate that hypoxia suppresses type-I interferon. To strengthen the claim, the authors should also try a HIF-1 inhibitor (e.g., PX478) on virus-infected cells under normoxia and hypoxia to monitor IFN genes.
2. In figure 4q, the authors show that IRF3-P10-OH has a temporary increase upon viral infection. To demonstrate the physiological function of EGLN-1 during viral infection the authors should further explore the potential mechanisms (e.g., due to an increased activity of

EGLN1 or due to a temporary complex formation between EGLN1 and IRF3).

3. In Figure 5, rather than ectopic expression of EGLN1-WT or H313A, complementing EGLN1 knockout cells with WT or H313A EGLN1 to monitor IRF3 dimerization and phosphorylation in viral infections would be better.

4. Please perform a quantification for IRF3 nuclear localization (% in nucleus) for Figure 5l.

5. The authors show that IRF3 P10A demonstrates impaired dimerization and phosphorylation, however, the exact mechanisms are not clear. Does P10A impact the binding of IRF3 to TBK-1/MAVS/STING/genome?

6. In Figure 6w and 6x, the authors show that PX478 treatment in vivo does not eliminate the difference between WT and P10A. Due to lacking non-treated controls it is not clear to me whether PX478 treatment has any effect on IFN. Could the authors utilize the BMDC extracted from IRF3 WT and P10A mice to perform ex vivo viral stimulation in the presence and absence of PX478?

7. In Figure 6, treatment of FG4592 further decreases the interferon responses in IRF3 P10A mice, suggesting that EGLN1 may have additional targets in the interferon signaling. Could the author comment on this?

Minor:

1. Line 46-50 needs further clarification. Do all viruses have selective tropisms for high-oxygen environments? It seems to be contradictory to the next sentence describing 'viruses that naturally infect organs with low oxygen tension'.

2. Line 53, 'on one hand' instead of 'on the one hand'.

3. The mass spec results are unreadable (Figure 4f, 4s and 4u).

4. Extended Data S3, figure legend for S3C mentions both SeV and VSV. Besides, please avoid using 'vs' and state clearly which part is the denominator.

Reviewer #1 (Remarks to the Author):

Upon sensing of intracellular viruses either via their DNA or RNA, signaling cascades are initiated that eventually converge in the activation of the transcription factor IRF3, which then drives expression of type I Interferons and other pro-inflammatory cytokines. The activity of the interferon induction cascade is tightly regulated and post-translational modifications (PTM) play a crucial role.

In this manuscript Xing Liu and coworkers show that hypoxic conditions suppress induction of antiviral genes independent of HIF signaling. They show that under normoxia the prolyl hydroxylase EGLN1 catalyzes the hydroxylation of IRF3 at proline 10. This PTM enhances IRF3's activation, leading to an enhanced gene activation under normoxia, which is suppressed under hypoxia. Molecular studies are elegantly combined with *in vivo* studies in mice and zebrafish, using *Egln1* deletion mutants that showed higher susceptibility to viral infection, confirming the *in vitro* phenotype and mechanism.

Overall, the data and conclusions of the manuscript are strong, relevant, highly interesting and convincing. The experiments are well conducted and controlled. Thus, the hypothesis is fully supported by the created data. I only have a few suggestions that may improve the current manuscript.

Major:

- The figures are a bit overbearing. I suggest to streamline the main figure panels focusing on the key information in the figures, and move e.g. repeats in different cell lines or alternative readouts to the supplements. This allows the reader to assess the data better and additionally facilitates the readability of figure information, which is currently occasionally very small. For example, the error bars, and individual points are barely discernable, e.g. Fig. 1b-d, f, g, i-k, m – q; Fig. 2b-d, f-l, k, m, o, q, r; Fig. 6b, d, k, g, h, i, j, q-x, axes in the flow cytometric analyses are very small (Fig. 2g, s), legends could be reassigned and images with increased resolution should be used in the following figures (e.g. in Fig. 3a the loxP site indicated green is hardly visible, b-m, o-q; Fig. 4f, s, u) and Fig. 3 and 5 containing *Egln1*^{-/-}, *ELGNI*^{-/-}, and *ELGNI*^{+/+}, *Egln1*^{+/+} and *Egln1*^{fl/fl}.

Response: Yes, we really appreciate the reviewer's suggestions. In this revision, we have reorganized and enlarged the figures and tried to increase the resolution of the figures. We have increased the number of main figures from 6 to 9 and moved some of the inner figures to extended data figures. Hopefully these changes will satisfy the reviewer this time.

- The authors nicely showed the detrimental effect of *Egln1*^{-/-} in mice. However, the viral load (e.g. via plaque forming assay or qPCR) during the infection/at the timepoint of sacrificing would provide valuable insight. I also suggest to quantify the immunofluorescence/flow cytometry viral loads in Fig. 2g, s, 3c. In addition, I would suggest to add a statement whether the baseline of inflammation i.e. in the non-infected animals were similar in KO and WT mice. Do KO mice already start with lower baseline restriction factor/ISG expression (e.g. as a supplementary figure).

Response: As suggested, we have reported the viral load at the time of sacrifice (Extended Data Fig. 6h in this revision).

Extended Data Fig. 6h.

Yes, we also have quantified the immunofluorescence/flow cytometry viral loads in Fig. 2g, s,3c. (Fig. 2b and 2 h, Fig. 3c in this revision).

Fig. 2b

Fig.2h

Fig.3c

C

Yes, we have also provided additional figures to show that the baseline of inflammation in the non-infected animals was similar in KO and WT mice (Extended Data Fig. 6d-6f in this revision).

Extended Data Fig. 6d-6f

- The discussion part could be more comprehensive. The authors could discuss their findings in a physiological or clinically relevant context, this would highlight the importance of the displayed results. Further, the relevance of the identified system e.g. during infection which causes hypoxia could be discussed and approaching the question why it is in place. Related to this, are there any other known oxidational processes in other immune pathways?

Response: Yes, we agree with the reviewer. In the revision, we have tried to discuss more related to our findings.

Minor:

- Check for abbreviation completeness (e.g. Fig. 3 (Cre-ER, ER = ?) and remove abbreviations where they are not necessary (e.g. Line 39-43).

Response: We have added (ER=Estrogen receptor) in figure legend and removed abbreviations where they are not necessary.

- Please maintain similar abbreviations throughout the manuscript e.g. Irf3 vs IRF3.

Response: Yes, I understand the reviewer's concern. Actually, Irf3 refers to mouse gene (*italic*) and protein, but IRF3 refers to human gene (*italic*) and protein.

- Fig. 3D: Move GFP band in the center of the depiction.

Response: Yes, we have revised accordingly (Extended Data Fig. 6c in this revision).

- The methods descriptions are brief and detail could be added. I further suggest to deposit large datasets on public databases instead of making them available only upon request (as currently stated in the manuscript).

Response: We understand the reviewer's concern. We have tried to expand the methods descriptions in this revision. In addition, we have deposit large datasets on public databases. Raw mass spectrometry data have been deposited to the ProteomeXchange Consortium (<http://proteomecentral.proteomexchange.org>) via iProX partner repository with the dataset identifier PXD048667. The original RNA-seq data were uploaded to the GEO Datasets (GEO accession GSE253181).

Reviewer #2 (Remarks to the Author):

Liu Nature Communications

This is an interesting paper linking oxygen availability to innate immunity via the hydroxylation of IRF3. Since the field is riddled with non-reproducible hydroxylation targets, it is imperative that the authors tighten up the evidence that IRF3 is hydroxylated in cells and by EglN1 *in vitro* (specific comments below). They would also be well served to delete or repeat (in some cases with advice below) some of the experiments where the differences observed were very small and hence of uncertain biological relevance.

Major Comments:

1. On a proportional basis the induction of IFN-beta by EglN1 compared with EV in the VSV treated cells in Extended Data Fig. 7C (*IRF3*^{-/-} cells) seems as great (or greater) than, for example, in the *IRF3*^{+/+} cells in Extended Data Fig. 1B. I understand the former and not the latter achieved statistical significance but that could simply reflect the noisiness of 7C (i.e. a few more replicates and it might become significant).

Response: Yes, we fully understand the reviewer's concern. We have repeated the experiments in *IRF3*^{-/-} cells and obtained new data that are more convincing (Extended Data Fig. 10c in this revision).

2. The description of how Fig. 4F was done in Results and Legend is COMPLETELY inadequate. I can't even tell if it was an *in vivo* or *in vitro* assay. Overexpressed IRF3 or endogenous? I can't read the spectra (even after magnifying) and should include analysis of the corresponding synthetic peptides with P10 hydroxylated or not.

Response: We apologize for our unclear description of Figure 4F in the Results and Legend, which caused confusion for the reviewer. In fact, it was an *in vivo* assay, as we described in the Methods section. Flag-IRF3 was overexpressed in HEK293T cells. The cell lysate was then immunoprecipitated with anti-Flag Ab-conjugated agarose beads. Immunoprecipitated IRF3 proteins were subjected to an 8% SDS-PAGE gel, and IRF3 bands were excised from the gel and analyzed by mass spectrometry.

3. Fig. 4H can't be interpreted- it only says the antibody requires the proline. A better experiment would be plus or minus DMOG or plus or minus EglN.

Response: We fully understand the concerns of the reviewer. To be honest, this is kind of a pilot experiment, we just wanted to use the only commercially available anti-hydroxyproline antibody (Abcam, Cat# ab37067) to quickly test that P10 of IRF3 is hydroxylated, then we can develop a specific anti-IRF3-P10-OH antibody. In this experiment, we first used the anti-hydroxyproline antibody to perform an immunoprecipitation to pull down the proteins with the prolines that can be hydroxylated. In wild-type IRF3, it contains P10, so if the P10 can be hydroxylated, WT IRF3 should be pulled down by the anti-hydroxyproline antibody and then can be detected by the anti-IRF3 antibody. However, in the mutant IRF3 P10A there is no P10, so it cannot be hydroxylated and therefore cannot be pulled down by the anti-hydroxyproline antibody, making it undetectable by the anti-IRF3 antibody.

Yes, we totally agree with the reviewer that it would be better to repeat this experiment plus or minus DMOG or plus or minus EglN. As you can see, in this study we have continued to use the specific anti-hydroxylated IRF3 P10 antibody to perform all these types of experiments, plus or minus DMOG or plus or minus EglN (Figure 6f and h in this revision).

4. Fig. 4L is unacceptable/not believable (difference too subtle). Better to reintroduce wild-type or mutant EglN1 into EglN1^{-/-} cells to get better signal to noise.

Response: Yes, we have followed the reviewer's instructions to repeat this experiment. We used EglN1^{-/-} H1299 cells instead of EglN1^{+/+} HEK293T cells to perform this experiment and obtained more dramatic data (Fig. 6e in this revision).

Fig. 6e.

5. Fig. 4P-Q also very poor. I applaud the attempt to quantify but doesn't look robust or biologically relevant.

Response: Yes, we agree with the reviewer. In this revision, we have repeated this experiment and re-quantified the data (Fig. 6i and 6j in this revision).

Fig. 6i

Fig. 6j

6. The differences in Fig. 5I and 5K are completely underwhelming.

Response: Yes, we fully understand the reviewer's concern. In this revision, we have repeated these experiments and obtained new data (Fig. 7f and Extended Data Fig. 12d in this revision).

Fig.7f

d

7. I can't read the spectra (even after magnifying) in Fig. 4U and should include analysis of the corresponding synthetic peptides with P10 hydroxylated or not. For Figs. 4F and 4U might also be helpful to comment on percent hydroxylation.

Response: We apologize for the confusion caused to the reviewer by the relatively smaller spectrum figure. Due to space limitations, we were unable to provide the larger spectrum figure in the manuscript previously. In this revision we have tried to provide the relatively larger spectrum figure of the original Figure 4U (Figure 6m in this revision). We hope that this will help the reviewer to read the spectra.

For Figs. 4F and 4U, following the reviewer's suggestion, we have provided the percentage hydroxylation of Figs.4F and 4U in figure legend (Figs 6a (5.56%) and 6m (10.34%) in this revision).

Fig.6m

m

8. The differences in Fig 6L and 6M are completely underwhelming.

Response: We fully understand the reviewer’s concern. In this revision, we have repeated these experiments and obtained new data (Fig. 9h and j in this revision).

Fig. 9h and 9j

h

k

j

l

9. A caveat is that the proline to alanine mutant is a non-conservative mutation that could affect function independently of hydroxylation. The use of FG4592 and *EglN1*^{-/-} cells is very helpful, however,

Response: Yes, we completely agree with the reviewer. As you can see, in this study, in addition to using the proline-to-alanine mutant, we also use FG4592 and *EglN1*^{-/-} cells for assays.

Minor Comments:

1. Line 77. Make no sense to write “However, how the hydroxylation of IRF3 function...” here because the authors haven’t yet described the fact that IRF3 is hydroxylated.

Response: Yes, we agree with the reviewer. In this revision, we have changed this sentence.

2. I think the Fig. 1 legend should give a sense of how much VSV and how much poly I:C was added.

Response: Yes, we agree with the reviewer. In this revision, we have provided the amount of VSV (+, 0.2 MOI; ++, 0.5 MOI; +++, 1.0 MOI; +++++, 2.0 MOI) and poly I:C (+, 0.5 µg/mL ; ++, 1.0 µg/mL) in figure legend.

3. Line 96. I think the authors mean “consistent with these findings”, not “consistently”

Response: Yes, the reviewer is right. We have changed that in this revision.

4. Line 107. I think the authors might mean “rarely”, not “barely” and might want to be saying “can be” less than 1%.

Response: Yes, the reviewer is right. In this revision, we have changed "barely" to "rarely" accordingly. What we really want to say is "cannot be" less than 1% under physiological conditions.

5. Line 114. Should just describe the finding. “cope with the decrease” infers the biological rationale, which doesn’t really belong here.

Response: Yes, we agree with the reviewer. We have changed the sentence in this revision.

6. How do the levels of exogenous *EglN1* in Fig. 2A compare with endogenous levels?

Response: In this revision, we have provided the levels of exogenous and endogenous *EglN1* in Fig.2A (Extended Data Fig. 2a in this revision).

7. The induction of IFN β and CXCL10 in extended data Fig. 1 is very modest (maybe because EglN1 is not limiting in these cells) and of uncertain biological relevance.

Response: In this revision, we have repeated these experiments and provided new data (Extended Data Figs. 2e and 2f in this revision), which are more dramatic than the previous data.

Extended Data Fig. 2e and 2f

8. Line 135, etc. Once again, “consistently” is not the right word here (consistently could mean reproducibly here). Think the authors mean, consistent with these findings”

Response: Yes, we agree with the reviewer. In this revision, we have changed accordingly.

9. The Figure 2 legend should make clear whether the knockout cells are clonal or polyclonal.

Response: Yes, as suggested, we have added “*EGLN1*^{-/-} H1299 cells are clonal” to the figure legend.

10. Why introduce IFIT1 measurements in Fig 2O? And the effect of wild-type EglN1, although statistically significant, is unlikely to be biologically relevant.

Response: As suggested, we have removed this figure in this revision.

11. Should acknowledge that this is not the first report of conditional EglN1 mice.

Response: Yes, we agree with the reviewer that this is indeed not the first report of conditional EglN1 mice. However, this mouse line was completely generated by us using a commercial company's service, we did not obtain it from other labs.

12. HIF stabilization usually lowers protein translation. How do you know the effect in Extended Data Fig. S5H is specific?

Response: We fully understand the reviewer's concern. Yes, HIF stabilization usually lowers protein translation. To address this issue, in this revision, we have used another transgenic zebrafish line, Tg (lyz: DsRed2) (in which the fluorescent protein, DsRed2, is driven by the zebrafish lyz promoter) for further assays. This fish line can be used to monitor myeloid precursors expressing lysozyme c gene (lyz) as well as neutrophils in the blood. However, compared to the control (DMSO), treatment with FG4592 had no apparent effect on the changes in red fluorescence signaling in SVCV-infected zebrafish larvae (Extended Data Fig. 8e, f). These data further suggest that FG4592 may specifically affect SVCV-induced IFN activation.

Extended Data Fig. 8e, f

13. A caveat with Fig. 4A-C, of course, is the proteins are OVERproduced. Are the results at least specific to EglN1 relative to EglN2 and 3?

Response: To address this concern, in this revision, we performed co-immunoprecipitation assay including EGLN1, EGLN2 and EGLN3 simultaneously. The result shows that only ectopically expressed EGLN1, but not EGLN2 or EGLN3, could pull down ectopically expressed IRF3 in HEK293T cells (Extended Data Fig. 9e in this revision).

Extended Data Fig. 9e.

e

14. The description of Fig. 4E in the legend is inadequate (e.g. I guess the cells were also transfected to produce FLAG-IRF3 but that is not stated).

Response: We apologize for the incorrect description of the PLA assay in the figure legend and method. In fact, we only transfected HA-EGLN1 into H1299 cells and used anti-IRF3 antibody to detect endogenous IRF3. We have corrected this in this revision.

15. Extended Data S6A lacks specificity controls or information about protein concentrations.

Response: We apologize for the loss of the GST protein control marker in the figure, which caused confusion for the reviewer. In this revision, we have added the GST control marker to the figure (Extended Data Fig. 9b in this revision). The amount of GST, GST-IRF3 and His-EGLN1 was 1.0 μ g each.

Extended Data Fig. 9a.

b

16. The enhancement of IRF3 activity by EglN1 is very modest in Extended Data Fig. 8i. Wonder if would be more impressive in *EglN1*^{-/-} cells.

Response: We really appreciate this valuable suggestion. In this revision, following the guidance, we disrupted EGLN1 in H1299 cells instead of overexpressing EGLN1 to perform further assays. Disruption of EGLN1 in H1299 cells does indeed suppress WT IRF3-induced expression of *IFN-β*, *CXCL10* and *ISG15* mRNA, but has no effect on P10A IRF3 function (Extended Data Fig. 13f in this revision).

Extended Data Fig. 13f

f

17. Fig. 6W and 6X are simply “negative data” without real controls to show HIF is downregulated and with a very questionable tool compound. I would probably delete.

Response: Yes, we agree with the reviewer. In this revision, we deleted Fig.6W and 6X.

Reviewer #3 (Remarks to the Author):

In this manuscript, Liu et al. discover that hypoxia condition suppresses innate immune signaling pathways. Mechanistically, they report that the suppression is EGLN1-dependent and HIF-1-independent. EGLN1 catalyzes proline 10 hydroxylation of IRF3 which enhances IRF3 phosphorylation and dimerization, in the presence of oxygen. The authors further provide *in vivo* data with recombinant mice/zebrafish and EGLN inhibitor to consolidate the positive regulation on innate immunity by EGLN1. Overall, the study is thoughtfully designed with convincing data. There are several aspects that need further explanation/exploration to substantiate the conclusions:

Major:

1. In figure 1, the only data supporting the HIF signaling-independence are Figure 1e, 1f, and 1g, while the other data merely demonstrate that hypoxia suppresses type-I interferon. To strengthen the claim, the authors should also try a HIF-1 inhibitor (e.g., PX478) on virus-infected cells under normoxia and hypoxia to monitor IFN genes.

Response: Yes, we agree with the reviewer. In this revision, following the reviewer's suggestion, we used PX478, a specific HIF1 α inhibitor, to block HIF1 α activity in H1299 cells. As expected, the expression of *GLUT1* was not induced under hypoxia (extended data Fig. 1e in this revision), confirming the effective inhibition of HIF1 α by PX478 in H1299 cells. Upon VSV infection, the expression of *IFN- β* and *CXCL10* was still reduced under hypoxia compared to normoxia (Figure 1d in this revision), further confirming the independence of HIF signaling.

Extended Data Fig. 1e.

e

Fig. 1d

2. In figure 4q, the authors show that IRF3-P10-OH has a temporary increase upon viral infection. To demonstrate the physiological function of EGLN-1 during viral infection the authors should further explore the potential mechanisms (e.g., due to an increased activity of EGLN1 or due to a temporary complex formation between EGLN1 and IRF3).

Response: To be honest, this is a very valuable idea that prompted us to further investigate the underlying mechanism. In this revision, we first performed a co-immunoprecipitation assay between IRF3 and EGLN1 during VSV infection. Indeed, the interaction between IRF3 and EGLN1 was stable during VSV infection (Extended Data Figure 11d in this revision). We then examined whether the hydroxylase activity of EGLN1 varied during VSV infection, using well-defined hydroxylated HIF1 α as an indicator. The results showed that the hydroxylase activity of EGLN1 was increased upon VSV infection for 2 h, but then decreased upon VSV infection for 4 h (Extended Data Figure 11e, f). Therefore, it appears that the dynamics of IRF3 hydroxylation during viral infection is due to changes in the hydroxylase activity of EGLN1 during viral infection. To further validate the dynamics of IRF3 hydroxylation during viral infection, we repeated this experiment in this revision and obtained new data that are similar to the original data (Figure 6i and 6j in this revision).

Extended Data Figure 11d.

Extended Data Figure 11e and 11f.

f
Fig. 6i and 6j.

i**j**
3. In Figure 5, rather than ectopic expression of EGLN1-WT or H313A, complementing

EGLN1 knockout cells with WT or H313A EGLN1 to monitor IRF3 dimerization and phosphorylation in viral infections would be better.

Response: Yes, we agree with the reviewer. In this revision, following the guidance, we have performed this experiment in EGLN1 knockout cells and obtained new data (Fig.7f in this revision).

Fig.7f.

4. Please perform a quantification for IRF3 nuclear localization (% in nucleus) for Figure 5l.

Response: Yes, we have revised accordingly (Fig.7g in this revision).

Fig.7g

5. The authors show that IRF3 P10A demonstrates impaired dimerization and phosphorylation, however, the exact mechanisms are not clear. Does P10A impact the binding of IRF3 to TBK-1/MAVS/STING/genome?

Response: To address this issue, we performed a co-immunoprecipitation assay (Extended Data Figure 13h in this revision). Indeed, upon VSV infection, WT IRF3 pulled down more endogenous TBK1 than P10A IRF3, suggesting that IRF3 hydroxylation at proline 10 may promote the interaction between IRF3 and TBK1.

Extended Data Fig. 13h

h

6. In Figure 6w and 6x, the authors show that PX478 treatment in vivo does not eliminate the difference between WT and P10A. Due to lacking non-treated controls It is not clear to me whether PX478 treatment has any effect on IFN. Could the authors utilize the BMDC extracted from IRF3 WT and P10A mice to perform ex vivo viral stimulation in the presence and absence of PX478?

Response: Following this suggestion, we have performed these experiments (Extended Data Fig. 15j in this revision). Indeed, VSV-induced *Ifn- β* and *Ifit1* expression was still lower in *Irf3_P10A* BMDCs than in WT BMDCs in the presence and absence of PX478 treatment, suggesting that HIF1 α does not mediate the difference between WT and *Irf3_P10A* mice in response to viral infection.

j

7. In Figure 6, treatment of FG4592 further decreases the interferon responses in IRF3 P10A mice, suggesting that EGLN1 may have additional targets in the interferon signaling. Could the author comment on this?

Response: Yes, we also see that treatment with FG4592 can somewhat reduce the interferon responses in IRF3 P10A mice, but not always in a statistically significant way. Of course, as there are several molecules upstream of IRF3 in the interferon signaling pathway, we cannot rule out the possibility that other molecules in the interferon signaling pathway may be targeted by EGLN1. Further investigation of this possibility will improve our understanding of the function of EGLN1-mediated hydroxylation in interferon signaling. In fact, we commented this in Discussion part before.

Minor:

1. Line 46-50 needs further clarification. Do all viruses have selective tropisms for high-oxygen environments? It seems to be contradictory to the next sentence describing ‘viruses that naturally infect organs with low oxygen tension’.

Response: In this revision we have tried to clarify this issue. But it is really difficult. To be honest, it is actually contradictory in the field. In fact, not all viruses have selective tropisms for high oxygen environments, some viruses even have selective tropisms for low oxygen environments. So there is some conflicting literature in the field.

2. Line 53, ‘on one hand’ instead of ‘on the one hand’.

Response: We have revised accordingly.

3. The mass spec results are unreadable (Figure 4f, 4s and 4u).

Response: In this revision, we have tried to provide the bigger figures of mass spec results.

4. Extended Data S3, figure legend for S3C mentions both SeV and VSV. Besides, please avoid using ‘vs’ and state clearly which part is the denominator.

Response: We have made changes accordingly.

REVIEWERS' COMMENTS

Reviewer #1 (Remarks to the Author):

The authors have addressed all my comments. Combined with the changes requested by the other reviewers the current manuscript is now significantly stronger.

Reviewer #2 (Remarks to the Author):

The authors are to be applauded because they have done an excellent job improving their manuscript, both in terms of style and substance. It reads much better and the data are more compelling. A couple minor things they might consider:

1. I would delete lines 456-458 from the Discussion. The authors provide compelling evidence that IRF3 links EglN1 to innate immunity. Of course it is formally possible there are other EglN1 substrates linked to innate immunity, but I don't see any reason to invoke that possibility here (and it violates the law of parsimony).

2. I would also consider deleting or rewriting lines 451-454. Hypoxia certainly has multiple effects in vitro and in vivo. Yes the effects of hypoxia on innate immune responses in vivo are likely to be multifactorial, but I don't see why they are "likely to produce biased results". I don't believe this paper needs such hypoxia experiments, but it would have been technically feasible to make mice mildly hypoxic and to test the effects of hypoxia on IRF3 and innate immune responses.

3. I wonder if larger versions of Fig. 6A and 6M can be provided as Extended Data so they are easily read.

Reviewer #3 (Remarks to the Author):

The revised manuscript has appropriately addressed my concerns with new data and clarifications. I have no further comment.

Reviewer #1 (Remarks to the Author):

The authors have addressed all my comments. Combined with the changes requested by the other reviewers the current manuscript is now significantly stronger.

Response: We appreciate your thorough review.

Reviewer #2 (Remarks to the Author):

The authors are to be applauded because they have done an excellent job improving their manuscript, both in terms of style and substance. It reads much better and the data are more compelling. A couple minor things they might consider:

1. I would delete lines 456-458 from the Discussion. The authors provide compelling evidence that IRF3 links EglN1 to innate immunity. Of course, it is formally possible there are other EglN1 substrates linked to innate immunity, but I don't see any reason to invoke that possibility here (and it violates the law of parsimony).

Response: Yes, we agree with your comments. As suggested, we have removed lines 456-458 from the discussion.

2. I would also consider deleting or rewriting lines 451-454. Hypoxia certainly has multiple effects *in vitro* and *in vivo*. Yes, the effects of hypoxia on innate immune responses *in vivo* are likely to be multifactorial, but I don't see why they are "likely to produce biased results". I don't believe this paper needs such hypoxia experiments, but it would have been technically feasible to make mice mildly hypoxic and to test the effects of hypoxia on IRF3 and innate immune responses.

Response: Yes, we agree with your comments. As suggested, we have removed lines 451-454 from the discussion.

3. I wonder if larger versions of Fig. 6A and 6M can be provided as Extended Data so they are easily read.

Response: As suggested, we have provided larger versions of Fig.6A and 6M in the Source Data.

Reviewer #3 (Remarks to the Author):

The revised manuscript has appropriately addressed my concerns with new data and clarifications. I have no further comment.

Response: We appreciate your thorough review.